



# Understanding Changes in Iceland's Streamflow Dynamics in Response to Climate Change

Hordur B. Helgason[1,2], Andri Gunnarsson[2], Óli G. B. Sveinsson[2], Bart Nijssen[1]

[1]Department of Civil and Environmental Engineering, University of Washington, Seattle, USA

[2]Hydropower Division, Landsvirkjun, Reykjavík, Iceland

*Correspondence to*: Hordur B. Helgason (helgason@uw.edu)

**Abstract.** The hydrological cycle in high-latitude regions is undergoing significant changes due to climate change. Iceland, with its extensive data from undisturbed catchments, provides a unique opportunity to study these changes in snow- and glacier-melt dominated regions. The country's heavy reliance on hydropower, without any connections to other electricity markets,

makes understanding these changes crucial. Recent decades have seen warming outpace global warming trends in Iceland, along with increased precipitation, reduced glacier mass, rising soil temperatures and expanded vegetation cover. The impacts of these environmental shifts on streamflow remain largely unexplored. Our study uses the LamaH-Ice dataset, which includes streamflow observations, atmospheric forcings from climate reanalyses, and catchment characteristics, to investigate changes in Iceland's streamflow dynamics over recent decades. We first examine the long-term variability in streamflow and its primary

drivers, correlating it with major climate indices. We then analyze trends during the last 30 and 50 years in annual, seasonal, and daily streamflow volumes, the timing of the spring freshet, and extreme flow conditions, linking these changes to environmental conditions and catchment attributes. Results show high inter-annual variability, decadal fluctuations, and strong correlations with the Arctic Oscillation, as reported in earlier studies. Streamflow trends vary by location and river type, with increased precipitation driving higher annual average flows in most rivers, while summer flows have decreased in most rivers,

which is linked to lower summer precipitation and increased evapotranspiration. This study is the first to report coherent regional and seasonal trends in Icelandic streamflow. Annual low flows have increased in most rivers. Glacial rivers show positive streamflow trends during the last 50 years, but negative trends during the last 30 years. The findings offer crucial insights into Iceland's hydrological changes amid rapid climatic shifts, with broader implications for reservoir operations and water resources management. This study enhances our understanding of Icelandic hydrology and contributes to global

knowledge on climate-induced hydrological changes.

## 1 Introduction

Anthropogenic climate warming, primarily driven by greenhouse gas emissions, has caused widespread environmental changes around the globe (IPCC, 2023). The Arctic has experienced particularly profound effects, with warming occurring at close to four times the rate of the global average between 1979 and 2021 (Rantanen et al., 2022). This has led to an intensification of

the hydrological cycle in the region (Box et al., 2019; Rawlins and Karmalkar, 2024).



In Iceland, the average warming rate between 1980-2015 was 0.47°C/decade (Björnsson et al., 2018). Despite this significant warming, analyses of snow observations in Iceland have shown a significant increase in snow cover and snow depth in some regions of Iceland over the periods 1930-2021 and 2001-2021 (Eythorsson et al., 2023). This is attributed to increases in precipitation, as annual precipitation in Iceland increased by about 10% between 1980-2015, with substantial variations between locations (Björnsson et al., 2018).

In addition to changes in weather and snow conditions, various other environmental factors in Iceland have undergone significant changes that impact streamflow dynamics. Glaciers have lost 18% of their area and 16% of their mass since 1890, with the most rapid mass loss occurring between 1994 and 2010 (Aðalgeirsdóttir et al., 2020). Since 2010, the pace of glacier net mass loss has reduced (Aðalgeirsdóttir et al., 2020). Soil temperatures in Iceland have increased, soil frost depth and duration have decreased (Petersen and Berber, 2018; Zaqout et al., 2023; Zaqout and Andradóttir, 2023), and permafrost is warming at a high rate and showing signs of degradation, with evidence of disappearance in some areas (Etzelmüller et al., 2023). Since the 1980s, increases in vegetation cover have been observed in the arctic, with increases particularly high in Iceland (Raynolds et al., 2015).

It is, however, still unclear how these changes have affected streamflow in Icelandic rivers. No recent comprehensive studies exist in the literature on how streamflow dynamics have changed in Iceland in the past decades. Jónsdóttir et al. (2006, 2008) performed trend analyses for two streamflow series for the period 1942-2002 and ten streamflow series for the period 1961-2000 to determine long-term changes in streamflow. Modest and statistically insignificant trends were found for mean annual and seasonal streamflow. For the longer period, a positive trend of 4% per decade in annual streamflow was observed for one of the two rivers (Jónsdóttir et al., 2006). For the shorter period, no trends were found for annual streamflow. However, two out of ten stations showed a 6-7% increase per decade for summer streamflow, which was attributed to colder temperatures in spring, delaying snowmelt into the summer. The magnitude of floods had a positive trend in the spring and a negative trend in the autumn, and spring floods showed a trend towards later timing, though these findings were generally not deemed statistically significant. A key takeaway was that despite a substantial increase in precipitation, no corresponding increase was found in the flow of non-glacial rivers (Jónsdóttir et al., 2006). Blöschl et al. (2017) examined the timing annual highest floods across Europe between 1960-2010 and showed that (spring) flooding in southwestern Iceland occurred later in the year, while flooding occurred earlier (or changed little) in the northeastern part of the country.

Crochet (2013) examined the sensitivity of ten river basins to climate variability by comparing streamflow patterns during cold and warm years, as well as wet and dry years, using data from 1971-2006. The analysis revealed that streamflow seasonality is highly sensitive to temperature increases, manifesting as reduced snowpack, earlier snowmelt, and greater streamflow in winter, with decreased flow in summer for non-glaciated catchments.

In this study, we investigate the multi-annual variability in Iceland's climate and analyze changes in streamflow, aiming to link these changes to shifts in hydro-meteorological drivers and changes in glacier extent and mass. Unlike previous studies that analyzed only a limited number of streamflow stations, our study leverages a significantly larger observational network. The data we use is from the "LArge-SaMple Data for Hydrology and Environmental Sciences for Iceland" (LamaH-Ice) dataset





(Helgason and Nijssen, 2024a) which provides streamflow measurements from an extensive network of mostly undisturbed watersheds, enabling a comprehensive study of changes in streamflow dynamics over recent decades due to climate change. We calculate trends for various climate and streamflow metrics while accounting for natural climate variability. Our research addresses two key questions: (1) What are the trends in annual, seasonal, and sub-seasonal streamflow in Iceland over recent decades? (2) What factors contributed to these changes?

## 70   2 Data and methods

### 2.1 Study area

Iceland, positioned in the North Atlantic, features a unique landscape shaped by glaciers that currently cover 10% of the country, active volcanism, and distinctive hydrological characteristics. Geologically, Iceland is bisected by a volcanic rift zone, leading to varied bedrock conditions that significantly affect hydrological patterns. Rivers originating from areas with porous,
young bedrock exhibit high baseflow, while those from regions with older, less permeable bedrock are primarily surface-fed. Despite the dominance of natural landscapes, with urban and agricultural areas comprising only a small fraction of the land, the hydrological system is complex. Rivers are categorized by their sources as glacial, direct-runoff, or spring-fed, although many receive contributions from multiple sources.

### 2.2 The hydrology and climate of Iceland

Iceland's hydrology is profoundly influenced by it's location, subject to frequent cyclones crossing the Atlantic from west to east and abundant precipitation, especially in winter. Iceland's climate is marked by high interannual variability, largely driven by large-scale atmospheric circulation patterns. The Arctic Oscillation (AO) plays a significant role, with a strong polar vortex (positive AO) trapping cold air in the Arctic and leading to milder, wetter conditions in Iceland (Thompson and Wallace, 1998). The Icelandic Low (IL), a semi-permanent low-pressure system between Iceland and Greenland, significantly shapes
the path of cyclones crossing the Atlantic. The North Atlantic Oscillation (NAO) index measures the pressure difference between the IL and the Azores High (Wanner et al., 2001). During a positive NAO phase, the IL intensifies, enhancing westerly winds that bring warmer, more humid air to Iceland, increasing precipitation, particularly in the south and west. Streamflow in Iceland from 1966 to 2004 has been shown to be correlated with the AO and the NAO, though the latter's influence is generally less significant (Jónsdóttir and Uvo, 2009). Studies have also identified a strong positive correlation between
streamflow magnitude and the intensity of southerly winds at the 500 hPa level (Snorrason, 1990). Additionally, sea surface temperatures (SSTs) around Iceland further modulate surface air temperatures, with warmer SSTs increasing glacial melt and runoff in glacial rivers (Jónsdóttir and Uvo, 2009).



## 2.3 Data

In this study, we use streamflow measurements, atmospheric reanalysis data and catchment characteristics from the LamaH-
Ice dataset (Helgason and Nijssen, 2024a). Most of the catchments in LamaH-Ice, or 79 out of 107 (74%), are unaffected by
anthropogenic influence such as flow regulations, water withdrawals or diversions. The dataset thus enables a comprehensive
study of changes in streamflow dynamics during the past decades due to climate change. We exclude streamflow data from
gauges heavily influenced by anthropogenic activities (e.g. hydropower withdrawals) or natural changes, such as evolving
watershed boundaries due to shifting river courses. However, gauges downstream of hydropower reservoirs are included in
the analysis of annual average streamflow trends when the reservoirs do not significantly affect annual streamflow volumes.
The locations of the streamflow gauges used in this study are shown in Fig. S22 in the Supplement, and Table S3 provides an
overview of the gauges, including river names, gauge locations, observation periods, and a selection of catchment attributes.
Streamflow measurements in Iceland are prone to interruptions (e.g., ice disturbances or instrument malfunctions), particularly
during winter, reducing data availability. For the analysis of trends in streamflow magnitudes, we did not fill gaps in the
records. We allowed up to 10% of daily data to be missing per year (or season) and calculated averages from the available data
without filling missing values. If more than 10% was missing, we excluded that year from the time series. We then calculated
trends for time series with at least 80% temporal coverage in annual values between the defined start and end dates. While
calculating trends for incomplete time series could bias the results, the inclusion of more gauges and regions enables a broader
and more representative analysis.

Meteorological time series in LamaH-Ice are derived from the ERA5-Land reanalysis (Muñoz-Sabater et al., 2021), which is
driven by the ERA5 reanalysis (Hersbach et al., 2020). ERA5-Land has a spatial resolution of 0.1°×0.1° (approximately 5 ×
11 km over Iceland). In this study, we use timeseries for total precipitation, snowfall, and temperature. The time series from
ERA5-Land are averaged over the watersheds upstream of each of the gauges. For watersheds with a significant portion of
glacierized areas, a substantial fraction of precipitation falling on the glacier accumulation zone is effectively disconnected
from immediate streamflow in the catchment. This is because such precipitation contributes to glacier mass, which only returns
as runoff up to decades later, creating a high degree of damping in the system.

Timeseries for total evaporation (ET) we use in this study are also from the ERA5-Land reanalysis. The reanalysis uses the
Carbon Hydrology-Tiled ECMWF Scheme for Surface Exchanges over Land (CHTESSEL) land surface model. Land
characteristics, such as glacier and vegetation cover, are represented by static masks based on satellite observations from the
1990s (Muñoz-Sabater et al., 2021). As a result, changes in glacier cover and vegetation are not reflected in ERA5-Land. The
maps in Sect. 3 use a basemap shapefile from Hijmans (2015) and glacier outlines from Hannesdóttir et al. (2020).

## 2.4 Homogeneity of streamflow series

Streamflow measurement series may exhibit inhomogeneity, meaning their statistical properties, such as mean or variance,
change over time due to factors like alterations in measurement practices or environmental conditions. To assess the




homogeneity of the streamflow records in LamaH-Ice, we performed the standard Petitt's test (Pettitt, 1979). Series identified
as inhomogeneous were manually inspected for breaks in homogeneity that were either 1) linked to a documented change in
measurement practices or to specific incidents that compromised data quality, or 2) distinctly observable in the data, and these
breaks could not be accounted for by shifts in temperature or precipitation. The homogeneity analysis revealed that one
timeseries needed to be omitted (Syðri-Bægisá river). The analysis, including the justification for each streamflow series, is

further described in the Supplement (Sect. S1). Our approach to considering or omitting inhomogeneous series aligns with that
of the Norwegian Water Resources and Energy Directorate's method for selecting reference streamflow series for climate
change studies (Fleig et al., 2013).

## 2.5 Calculation of spring freshet timing and centroid of timing

To calculate the timing of spring freshet and centroid of timing, complete and continuous series of streamflow were needed.
Many methods have been used in the literature to fill gaps in streamflow records, ranging from simple linear interpolation to
advanced statistical or hydrological modeling techniques (Dembélé et al., 2019). A commonly applied approach involves
interpolation from analogue gauging stations (WMO, 2008). Instead of relying on data from nearby gauges, we opted to fill
missing streamflow records using the inter-annual mean daily flow for the given gauge. This method leverages temporal
averaging and scaling based on observed streamflow conditions. Specifically, for each missing value, a 31-day window (15
days before and after the target date) was used to extract observed values. A scaling factor was then calculated as the ratio of
the median flow in this window to the mean daily flow over the same period. The missing value was subsequently estimated
by applying this scaling factor to the mean flow for the same day of the year from other years. We used the median flow in the
31-day window instead of the mean, to minimize the influence of extreme values. We applied this method for all gaps with a
duration of 60 days or less; longer gaps were not filled.

The centroid of timing was calculated as the day of the water year when 50% of the total annual flow volume has passed the
gauge. The timing of the onset of spring freshet was calculated using a method developed by Cayan et al. (2001). This method
tracks the accumulated difference from the average streamflow over a given water year. When the resulting curve reaches its
lowest point, it indicates that the spring freshet has begun, and the current streamflow magnitude has surpassed the annual
average. To pinpoint the actual onset of the freshet, we then identify the lowest point on the hydrograph from the preceding
days. Calculating a trend in these metrics provides insight into how the timing of spring snowmelt and the annual flow mass
has shifted over recent decades. The methodology was adopted from Berge et al. (2021).

## 2.6 Calculation of trends

Autocorrelation, which is often present in streamflow records, may influence the ability to detect trends (Yue et al., 2002). We
thus used the modified Mann-Kendall trend test (Hamed and Ramachandra Rao, 1998) to assess the significance of trends in
this study, with significance determined at $p < 0.05$. Unlike the traditional Mann-Kendall test, which assumes independence
among the observations, the modified version adjusts the variance of the test statistic to account for autocorrelation. We





calculated the magnitude of trends using the Theil-Sen (TS) estimator (Sen, 1968; Theil, 1950), which is calculated as the median of the slope of lines connecting all data point pairs. This trend estimation method is commonly used in hydrological studies as it is insensitive to outliers and suitable for skewed and heteroskedastic data.

We calculate trends for annual, seasonal and sub-seasonal averages. Seasons are defined as winter (December to February), spring (March to May), summer (June to August) and fall (September to November). For calculating sub-seasonal streamflow trends, we employ 21-day rolling mean (21DRM) centered on each day in the series, enabling us to determine a trend for each day of the year. This approach aligns with prior research, which has explored moving windows of varying durations, including 3-day (Kim and Jain, 2010), 10-day (Skålevåg and Vormoor, 2021) and 30-day periods (Kormann et al., 2015). Our selection

of a 21DRM balances the demand for a relatively high temporal precision against the challenges that arise as the averaging period decreases and data variability intensifies. For an in-depth explanation of the methodology, we direct readers to the study by Skålevåg and Vormoor (2021).

The high natural climatic variability in Iceland makes streamflow patterns and hydrological processes in Iceland highly dynamic, leading to significant fluctuations in precipitation, temperature, and runoff, on both an annual and decadal scale. As

a result, trend analysis in such a variable environment is highly sensitive to the period used. Shorter periods may capture trends that are not representative of longer-term changes, while long periods can obscure shorter-term fluctuations that are critical to understanding streamflow dynamics. This inherent variability complicates the detection of robust trends and the attribution of observed changes to specific climate drivers. To investigate the sensitivity of the time periods chosen and to leverage the availability of streamflow data in Iceland, we calculate trends for two time periods, 1973–2023 and 1993–2023. The earlier

period, 1973–2023, includes relatively few streamflow series (25), while more series extend back to 1993, allowing for a larger dataset in the 1993–2023 analysis (37 series).

## 3. Results

### 3.1 Multiannual variability in temperature, precipitation and streamflow

#### 3.1.1 Multiannual variability in temperature and precipitation

Since the late 19th century, Iceland has experienced significant warming, although this trend has not been continuous. A marked warming phase occurred during the second decade of the 20th century, followed by a cooling period that persisted until the late 1970s. From that point onward, rapid warming resumed and has continued into recent years (Björnsson et al., 2023). Figure 1 shows the annual average temperature and precipitation over Iceland since the mid-20th century, based on the ERA5-Land reanalysis. While temperature and precipitation vary across the country, the use of averages in Figure 1 helps

simplify and interpret large-scale, multiannual variability in Iceland's weather conditions. Because precipitation in the ERA5-Land reanalysis is underestimated for Iceland (Helgason and Nijssen, 2024a), we present it as percentage deviations from the mean rather than absolute amounts.

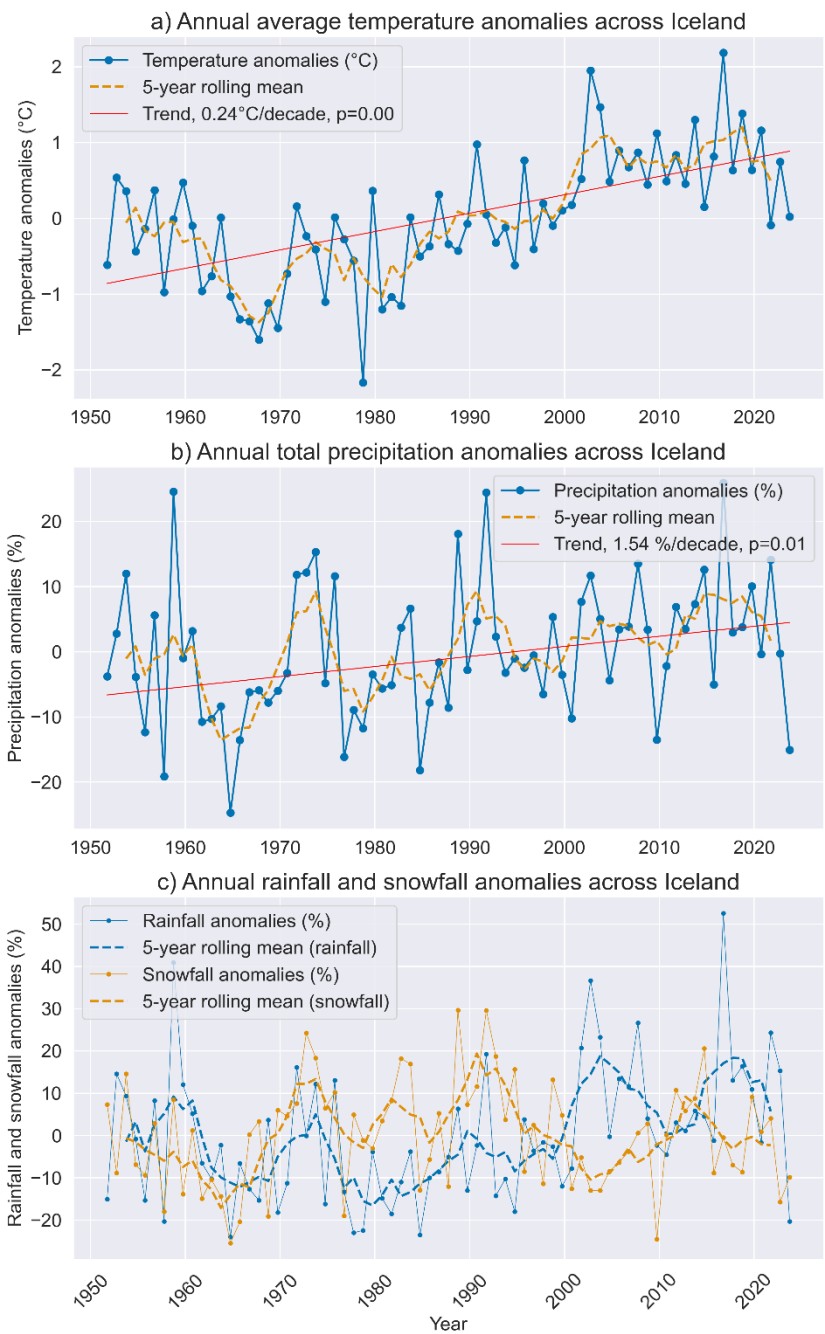

**Figure 1: Anomalies from the long-term mean (1951-2024) in annual average temperature (°C), total precipitation (m), and precipitation partitioned into rainfall and snowfall over Iceland, derived from the ERA5-Land reanalysis. Averages are calculated for water years from October 1, 1951, to September 30, 2024. Solid lines represent water-year averages with circular markers for each year, while dashed lines show the 5-year centered rolling mean. Trendlines for temperature and precipitation are calculated**




**using the Theil-Sen method, with statistical significance assessed using the Mann-Kendall test. Both trends are statistically significant (p < 0.05).**

The temperature data reveals a clear warming trend over this period, with a long-term increase of 0.24°C per decade (Figure 1a). A period of colder years is evident in the late 1960s, a time marked by substantial sea ice around Iceland, followed by a warming phase beginning in the 1980s. The 5-year rolling mean highlights this warming, with temperatures notably higher after 2000 compared to previous decades. The warming appears to have slowed in recent years. In the last three years (October 1, 2021, to September 30, 2024), two of the coldest years since the turn of the century were recorded (Figure 1a).

Precipitation over Iceland exhibits significant interannual variability, characterized by distinct peaks and troughs throughout the study period. Despite this variability, a statistically significant long-term upward trend of 1.54 % per decade is evident. Shorter periods of sustained high or low precipitation are visible in the 5-year rolling average, which often align with fluctuations in the smoothed temperature average.

The 2024 water year recorded the lowest precipitation since the 1980s, with only four other years in the past 74 years (1951-2024) showing lower values. This deviation from the overall upward trend shows that short-term dips in precipitation can still occur, even against a backdrop of long-term increases.

The partitioning of precipitation into rainfall and snowfall reveals that snowfall contributed significantly to total precipitation from around 1970 to 1995. From the early 2000s onwards, there is a shift towards more rainfall, with snowfall becoming less dominant. This shift coincides with the broader warming trend.

### 3.1.2 Multiannual variability in streamflow

Figure 2 illustrates the long-term variability in streamflow for Icelandic rivers with observational records dating back to before 1980. Distinct periods of streamflow variability are evident over the past several decades. The 1950s saw a high-flow period, followed by lower flows in the 1960s. Streamflow increased again during the 1970s, but the 1980s experienced another period of reduced flow. A brief high-flow period occurred in the early 1990s, after which streamflow declined until 2000. The 2000s marked a decade of high flows, particularly in glaciated catchments. Since the early 2010s, glaciated catchments have experienced near average or lower flows.



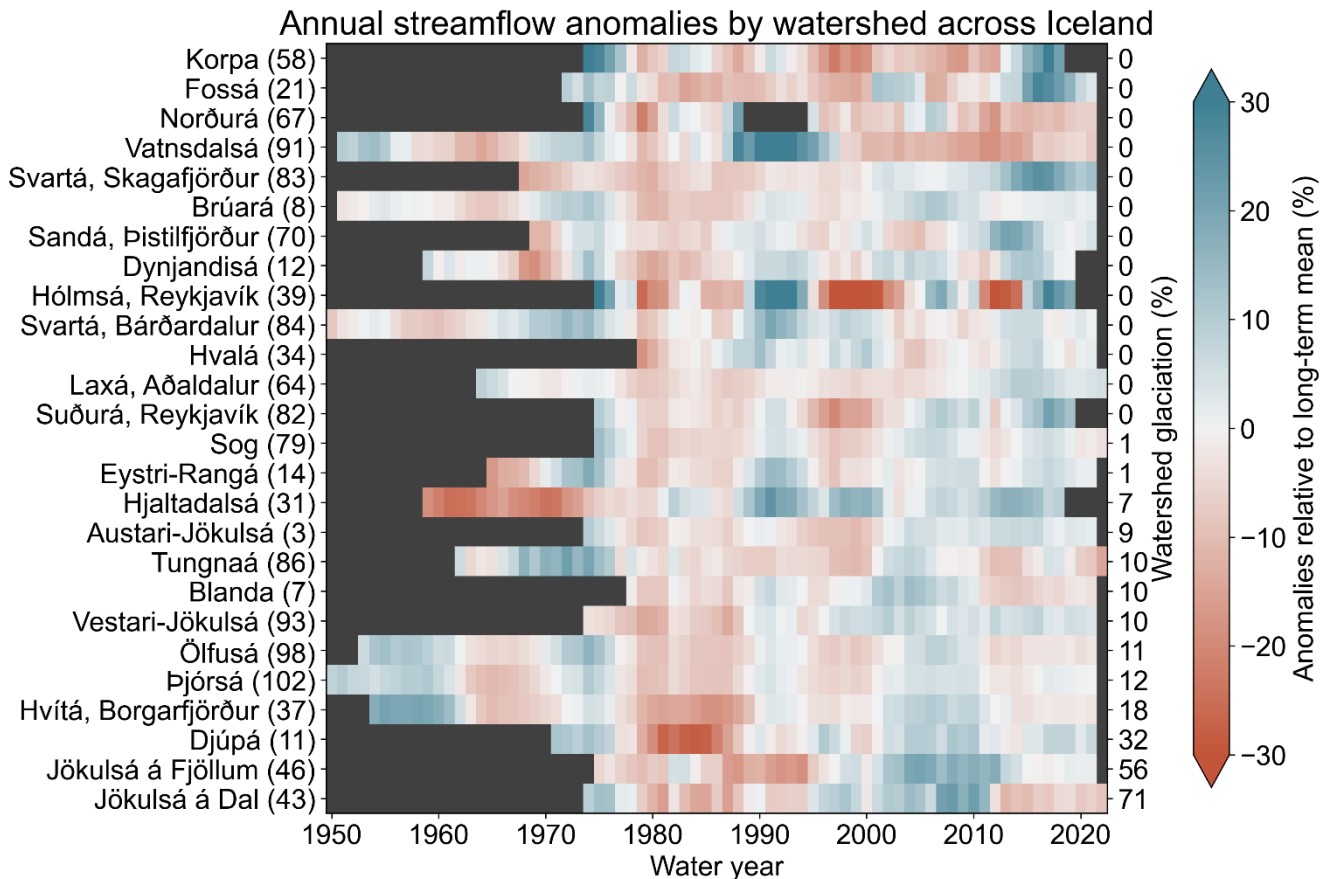

**Figure 2: Annual streamflow anomalies from the long-term mean for Icelandic gauges with extended measurement records. For each gauge, the long-term mean is calculated based on its available data, and yearly anomalies are expressed as percentages relative to this mean. A 5-year centered rolling average is applied to smooth short-term variability. The color scale indicates percentage anomalies, with red representing below-average flows and blue representing above-average flows. The left Y-axis lists river names and corresponding LamaH-Ice gauge IDs, while the right Y-axis displays watershed glaciation percentages.**

These streamflow patterns align closely with the temperature and precipitation trends shown in Figure 1. The high-flow periods in the 1950s and early 1990s coincide with increased precipitation (Figure 1b), while the elevated flows in glaciated catchments during the 2000s align with warmer temperatures (Figure 1a). Additionally, enhanced glacier melt in the 2000s was influenced by sub-glacial volcanic eruptions at Eyjafjallajökull (2010) and Grímsvötn (2011), with ash and tephra depositing on glaciers, reducing glacier albedo and increasing melt. Low-flow periods, such as in the 1960s, 1980s, and post-2010, correspond to reduced precipitation and/or lower temperatures.

### 3.1.3 Correlation of streamflow with climate indices



Figure 3 presents the correlation between water-year average streamflow in Icelandic rivers and the Arctic Oscillation (AO)
and North Atlantic Oscillation (NAO) climate indices. The AO and NAO indices were obtained from the NOAA Climate
Prediction Center (NOAA CPC, 2024). Streamflow with records dating back to before 1980 were used in the analysis (27
series). The streamflow series were normalized and de-trended before the correlation analysis. The climate indices, already in
a normalized form, were de-trended.

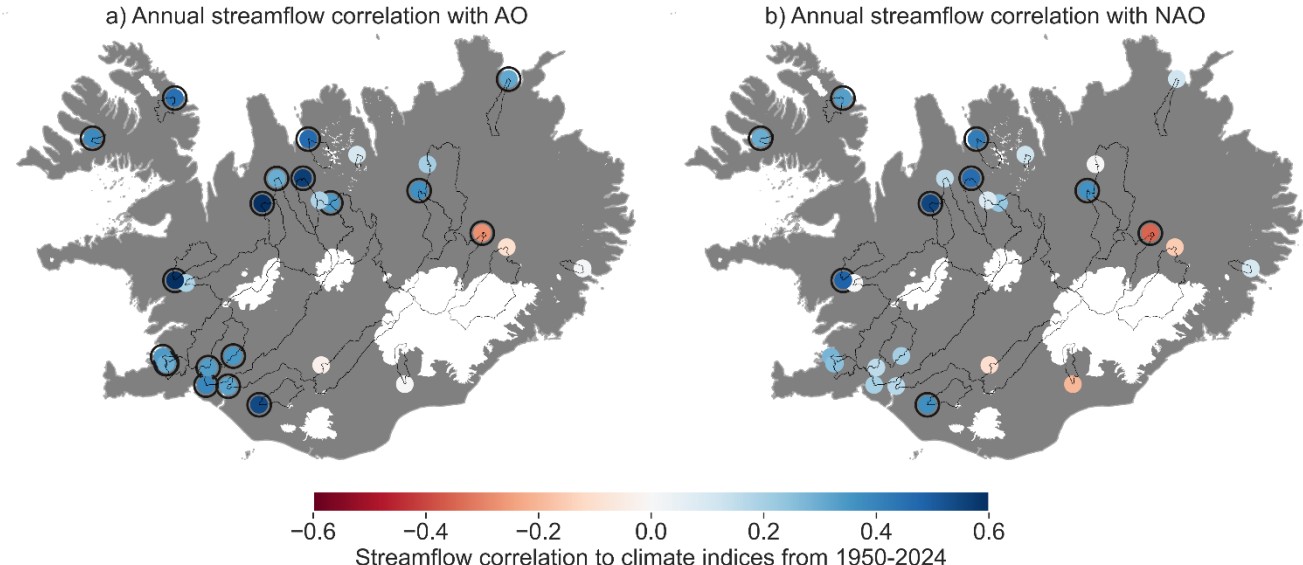


**Figure 3: Annual streamflow correlation with the Arctic Oscillation (AO) and North Atlantic Oscillation (NAO) climate indices for
streamflow gauges in Iceland. Black circles around gauges indicate statistically significant correlations (p < 0.05). The correlations
were computed using Pearson's method, with the time series de-trended and normalized beforehand. The analysis covers available
streamflow data, with the number of years in each series varying across 27 gauges. The average time series length is 58.7 years.**

Figure 3a highlights the strong influence of the Arctic Oscillation (AO) on streamflow variability in Iceland, with significant
positive correlations at 18 out of 27 gauges. In comparison, Figure 3b shows generally weaker but mostly positive correlations
between streamflow and the North Atlantic Oscillation (NAO), with significant positive correlations observed at 8 gauges.
This indicates that while the NAO's impact on streamflow is notable, it is less pronounced than that of the AO. These findings
align with the results reported by Jónsdóttir and Uvo (2009).

**3.2 Trends in temperature, precipitation and evapotranspiration**

We calculated annual and seasonal trends for catchment-averaged temperature, precipitation and evapotranspiration for all 107
catchments in LamaH-Ice. These timeseries are from the ERA5-Land reanalysis (Muñoz-Sabater et al., 2021).

**3.2.1 Annual trends**

Annual trends for the periods 1973-2023 (period 1) and 1993-2023 (period 2) are displayed in Figure 4.

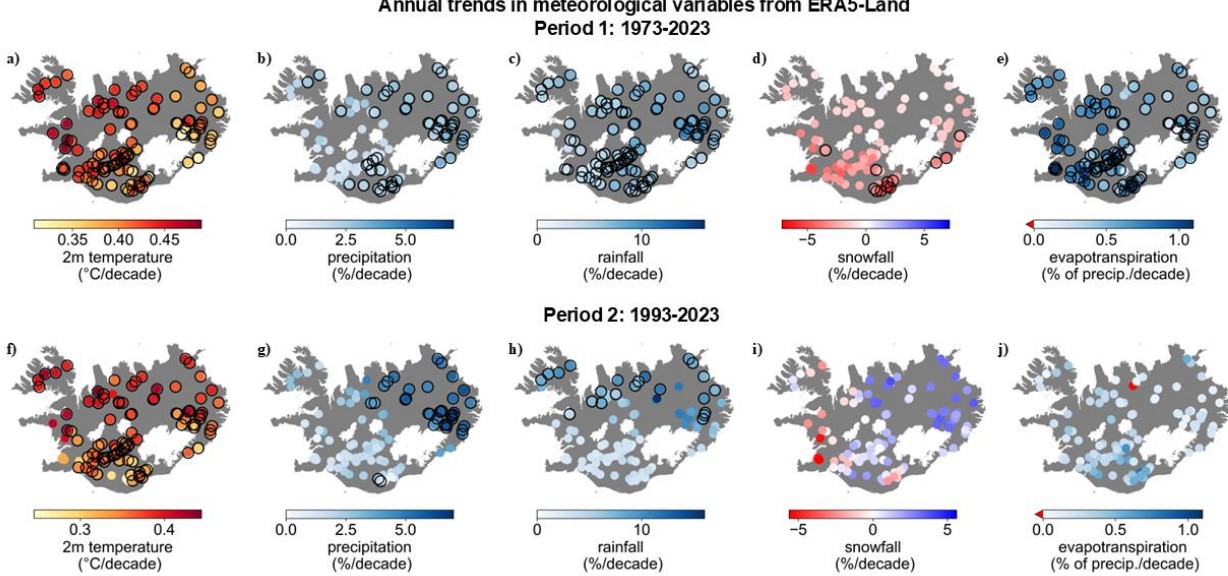

**Figure 4: Trends in catchment-average 2m air temperature (a and f), precipitation (b and g), rainfall (c and h), snowfall (d and i) and evapotranspiration (e and j) from 1973-2023 and 1993-2023, with each point marking the streamflow gauge location. Evapotranspiration trends are shown as percentage of annual precipitation per decade. Black circles around gauge markers indicate statistically significant trends (p < 0.05). The same figure is shown in the Supplement (Fig. S17), with precipitation and evapotranspiration in the units of mm/decade. The data is from the ERA5-Land reanalysis** (Muñoz-Sabater et al., 2021).

Figure 4a and f reveal significant increases in annual temperatures across all catchments in both periods, with stronger warming in the west, decreasing toward the east for period 1, and stronger warming in the north, decreasing towards the south for period 2. Precipitation trends are generally modest over these periods. In period 1, the strongest positive trends occur in the eastern part of the island (Figure 4b), and in period 2, trends are strongest in the northeast (Figure 4g). The magnitude of the trends is larger in period 2. Rainfall trends are positive and significant during period 1 (Figure 4c), but less pronounced in period 2 (Figure 4h). Snowfall displays near-zero or declining trends in all regions in period 1, with statistically significant decreases noted in the central south and southeast (Figure 4d). In period 2, snowfall increases are observed, although these trends are not statistically significant (Figure 4i). The strongest increases occur in the northeast, while decreases are observed in the west.

Evapotranspiration (ET) trends are generally slightly positive but vary across regions. In period 1, ET increases are more widespread (Figure 4e) and statistically significant in almost all locations, while in period 2, ET trends are lower and not significant (Figure 4j). The overall increase in ET aligns with rising temperatures. However, these increases are small, accounting for less than 1% of annual precipitation per decade in most catchments. Figure S18 in the Supplement shows a comparison between the magnitude of trends in precipitation and ET in each basin (further discussed in Sect. 3.3.1).





### 3.2.2 Seasonal trends

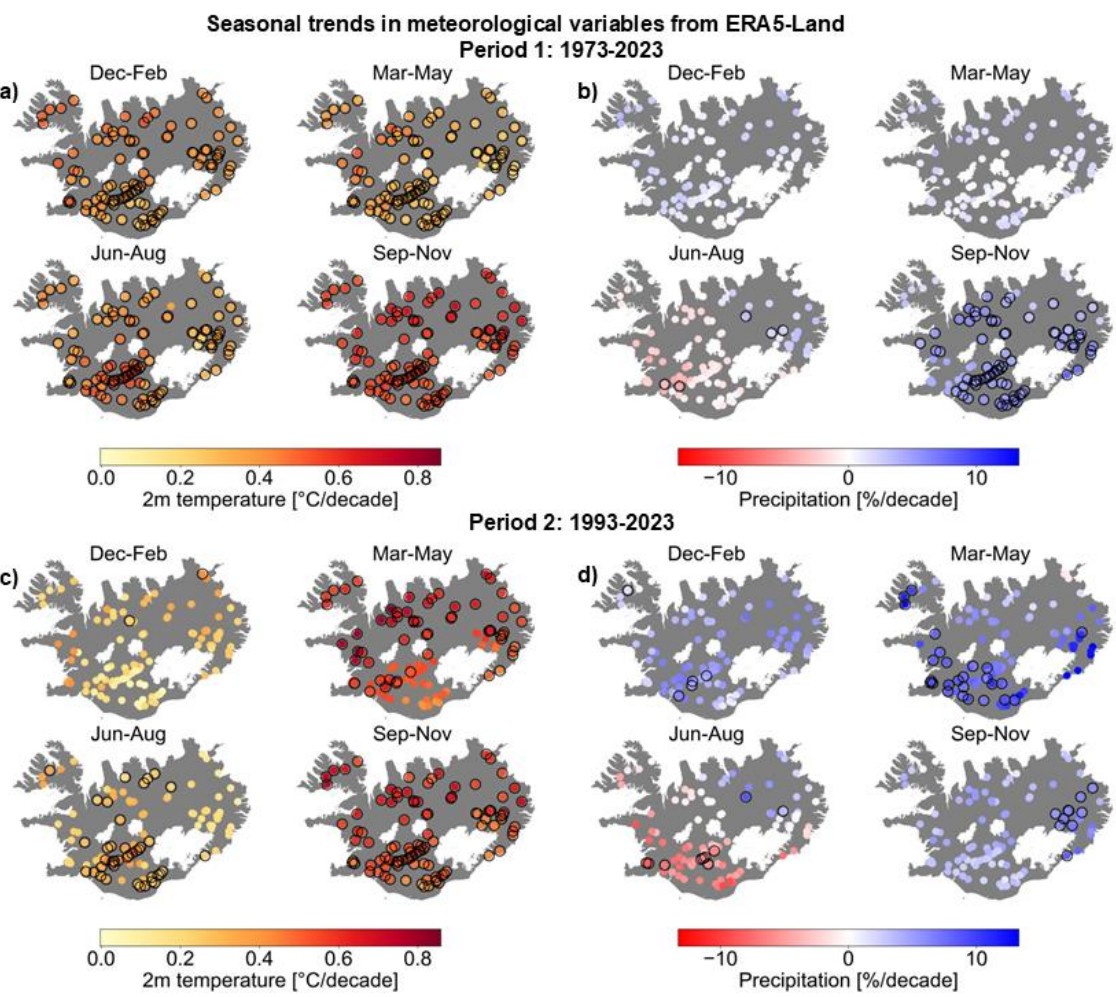

**Figure 5: Seasonal trends in catchment-average 2m air temperature (a and c) and precipitation (b and d) from 1973-2023 and 1993-2023, with each point marking the streamflow gauge location. Black circles around gauge markers indicate statistically significant trends (p < 0.05). The data is from the ERA5-Land reanalysis** (Muñoz-Sabater et al., 2021)**.**

Seasonal trends for 2m air temperature and precipitation for periods 1 and 2 are shown in Figure 5. The figure shows that, in almost all instances, temperature trends across all seasons for period 1 are statistically significant, displaying the highest increases in fall, whereas spring presents the smallest change. For precipitation trends (Figure 5b), significant positive trends are observed in the fall season for most catchments. The winter and spring seasons are characterized by a modest positive or near-zero trends. In the summer season, precipitation has slightly increased in the northeast region and decreased in the southwest.

Analysis of temperature trends for period 2 (Figure 5c) shows less statistical significance in seasonal temperature trends compared to period 1, with winter and summer now exhibiting the least pronounced trend, characterized by rather few



significant data points. Fall remains the season with the highest warming trend, but the spring season also has quite a high trend, which is a shift from period 1. As for precipitation (Figure 5d), the spring season shows the strongest positive trends in period 2, especially in the island's southwestern region. Winter and fall seasons exhibit positive trends, while summer persists
in presenting an overall downward trend for precipitation as in period 1, with highest decreases in the southwest.

### 3.2.3 Sub-seasonal trends in temperature and precipitation

**Figure 6: Sub-seasonal trends (21DRM) in catchment-average 2m air temperature (a and b) and precipitation (c and d) from 1973-2023 and 1993-2023. An overlying dashed black line indicates that the trend is significant (p < 0.05). Left Y-axis labels show the streamflow gauge ID number. Right Y-labels show the glaciated fraction of the catchment. The data is from the ERA5-Land reanalysis** (Muñoz-Sabater et al., 2021)**.**

Figure 6 shows the sub-seasonal trends in 2m air temperature and precipitation for periods 1 and 2. For air temperature, results for period 1 (Figure 6a) show a predominant significant positive trend in temperature for most of the year. A period in February and March exhibits the smallest (and not statistically significant) trends, along with two shorter periods in October/November
and December. Trends in July and August are also small in many catchments. In contrast, results for period 2 show greater



variability, including periods with negative trends (Figure 6b). Even though fewer days exhibit statistically significant trends than in period 1, the magnitude of the positive trends is notably larger. Significant increases in temperature are observed in February/March, late April and from late May to late June, while minor decreases are noted in August, December and January. The precipitation trends present mixed results. For period 1 (Figure 6c), a significant increase in precipitation is evident in

May and September and October across almost all catchments. Other parts of the year show limited significance, with slight increases in December and July, and decreases in March, June, August, and November. In period 2 (Figure 6d), statistically significant positive precipitation trends are observed in May and September in most catchments, with slight increases also noted in late December, March, April, late May and July. Decreases are noted in August and early December.

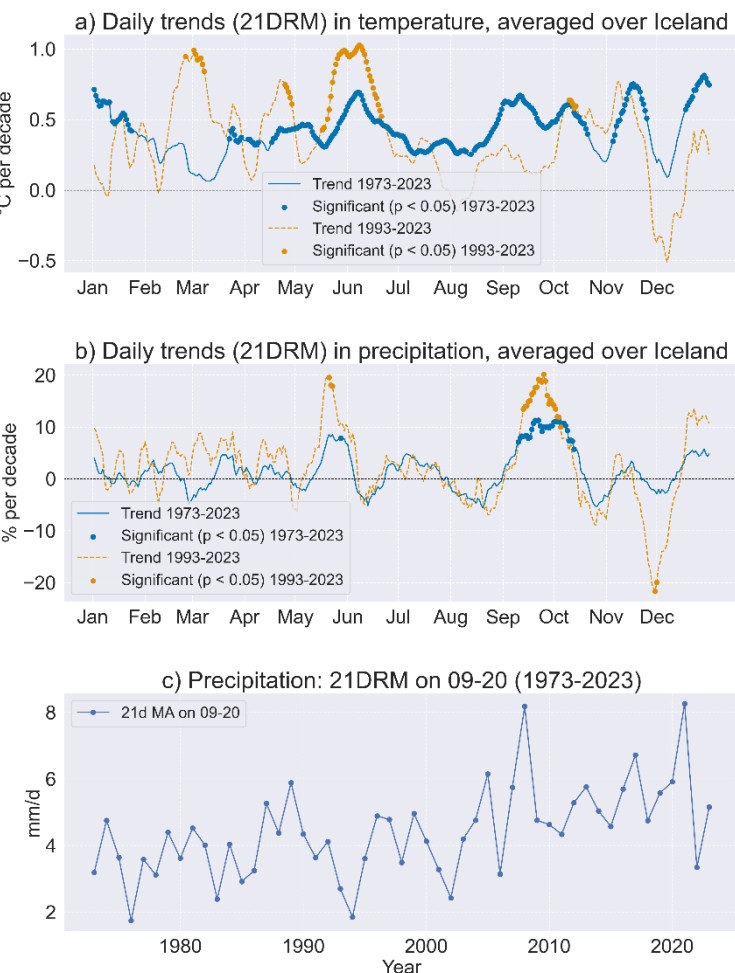

**Figure 7: Daily trends in 21-day rolling means of 2m air temperature (a) and precipitation (b) from the ERA5-Land reanalysis for Iceland, shown for the periods 1973 to 2023 (blue) and 1993 to 2023 (orange). Precipitation trends for September 20th are highlighted separately in panel (c).**



To better explain these variations within the year, Figure 7 shows average trend in 21DRM temperature and precipitation for all of Iceland for the two periods. The temperature trends in Figure 7a show that the values rarely fall below zero for either period. A notable exception for period 2 occurs in December, where decreasing trends are evident, corresponding to cooling during that month. Precipitation trends also show a similar pattern (Figure 7b), with a decrease in December for period 2, indicating concurrent cooling and drying. For temperature (Figure 7a), most of the year exhibits statistically significant increases for period 1. In contrast, statistically significant increases for period 2 are more limited, occurring mainly in two short periods in March and April and a more extended period from May to June. For precipitation (Figure 7b), statistically significant increases are noted in May for both periods, although only a few data points meet the threshold for significance. A consistent period of significant increases is observed in September and October. The daily precipitation values for September 20 (Figure 7c) confirm these trends, showing notable increases, particularly since the mid-2000s, emphasizing the validity of the observed upward trend in autumn precipitation.

### 3.3 Trends in streamflow

Annual, seasonal, and sub-seasonal trends were calculated for streamflow records in LamaH-Ice with sufficient data coverage (Sect. 2.3), for the two periods, in the same manner as for the meteorological variables described in Sect. 3.2. The trend values for each gauge are presented in Tables S1 and S2 in the Supplement.

### 3.3.1 Annual and seasonal averaged streamflow

Figure 8 shows the annual and seasonal trends in streamflow for the gauges with sufficient data coverage for the two periods. Figure 9 shows a summary of these results in a heatmap. Figure S20 in the Supplement shows trends in annual and summer melt season streamflow for glacial rivers only.





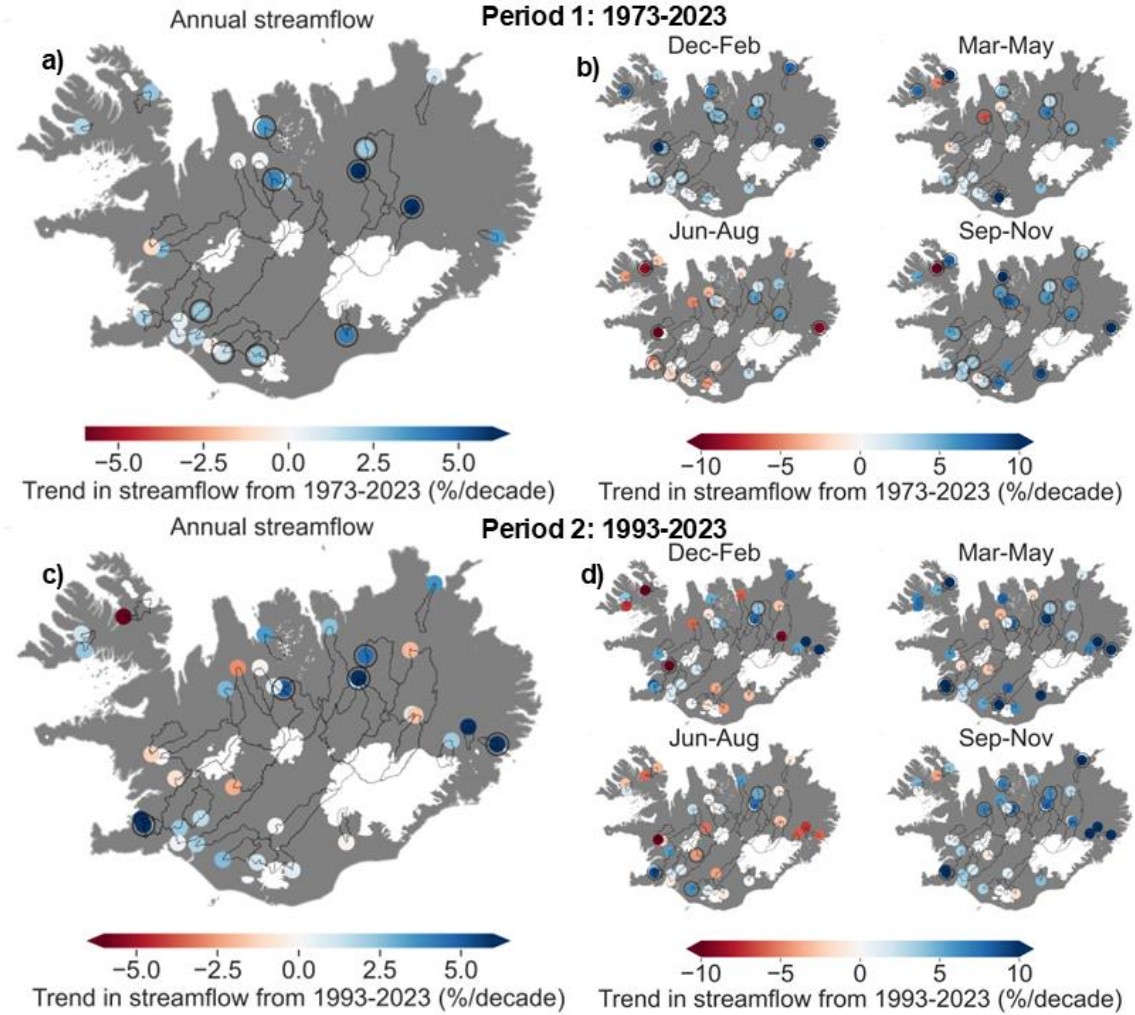

**Figure 8: Annual (a, c) and seasonal (b, d) trends in streamflow from 1973-2023 (a, b) and 1993-2023 (b, d). Black circles around gauge markers indicate statistically significant trends (p < 0.05). Watershed outlines are shown for each gauge.**




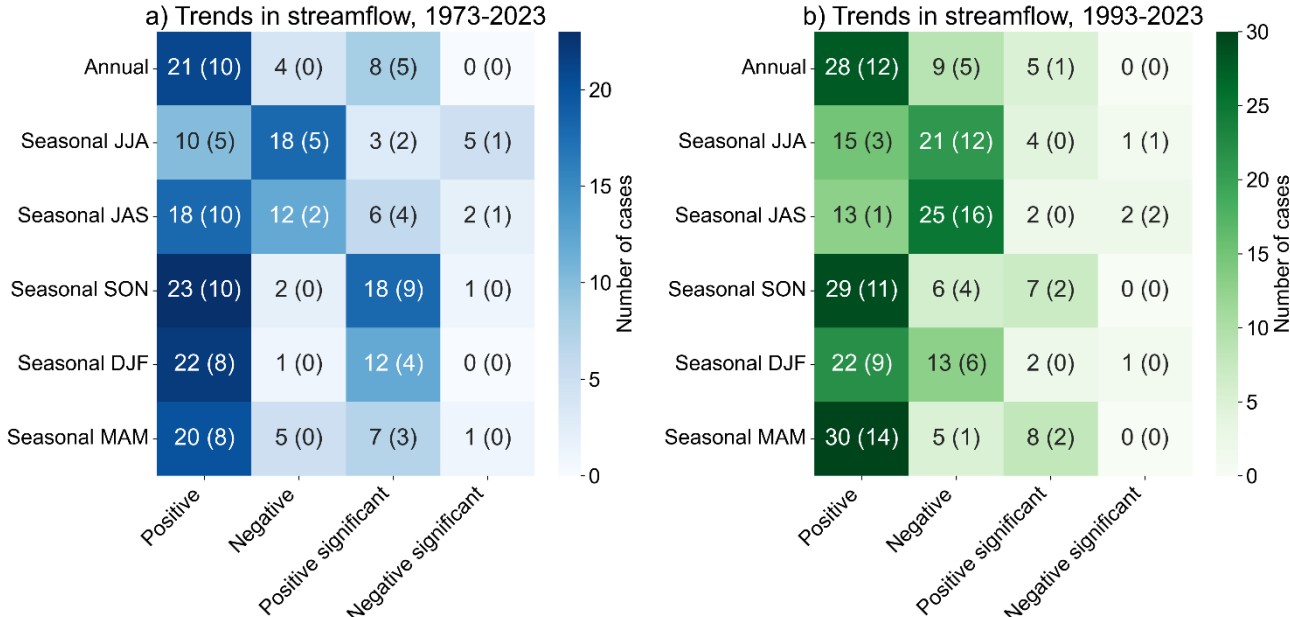

**Figure 9: Heatmaps showing a summary of the results from analysis of annual and seasonal trends in streamflow for the periods 1973-2023 (a) and 1993-2023 (b). The numbers in parentheses indicate the count of basins with more than 5% glaciation.**

*Period 1:*

Annual average streamflow for period 1 exhibits a positive trend in 21 out of 25 gauges (Figure 8a and Figure 9a). In fact, a positive trend is predominant in all seasons except summer. The gauges showing a significant increasing trend in annual average flow are in the northern part of the island (5 out of 11) and in the southern part (4 out of 14 - Figure 8a).

The seasonal analysis for period 1 (Figure 8b and Figure 9a) reveals that a majority of gauges show positive trends during the fall season, with 23 out of 25 gauges showing positive trends, thereof 18 gauges showing a statistically significant increasing trend. Positive trends are also found for winter season (22 out of 23 gauges), with 12 gauges showing statistically significant increasing trends, and spring seasons (20 out of 25 gauges), with 7 gauges showing statistically significant increasing trends. Conversely, summer (JJA) trends are negative in most gauges (18 out of 28), though only 3 of these negative trends reach statistical significance. The trend in annual average streamflow is positive for all glaciated basins in period 1, reaching significance in half of the gauges. The trend is positive during summer melt season (JAS) in most cases as well.

Figure S19 in the Supplement shows the streamflow, temperature and precipitation series for the 5 gauges in the northern part of Iceland showing a significant trend in annual average streamflow. The positive trends observed in the two rivers that drain the large ice-caps Hofsjökull and Vatnajökull, Vestari-Jökulsá (gauge ID 93) and Jökulsá á Fjöllum (46), can be attributed to several factors: (1) increased glacier melt driven by rising temperatures, (2) a glacier surge in the Dyngjujökull glacier, which feeds the Jökulsá á Fjöllum river, occurring in 2000 and increasing glacier area at lower elevations, thereby accelerating melt rates, and (3) volcanic eruptions in Eyjafjallajökull (2010) and Grímsvötn (2011), which lowered glacier albedo and further





enhanced melt. The positive trends in the remaining rivers, Hjaltadalsá (31), Laxá (64), and Sandá (70), are largely explained by a period of unusually high precipitation from 2012 to 2015, which significantly influenced annual streamflow during this period.

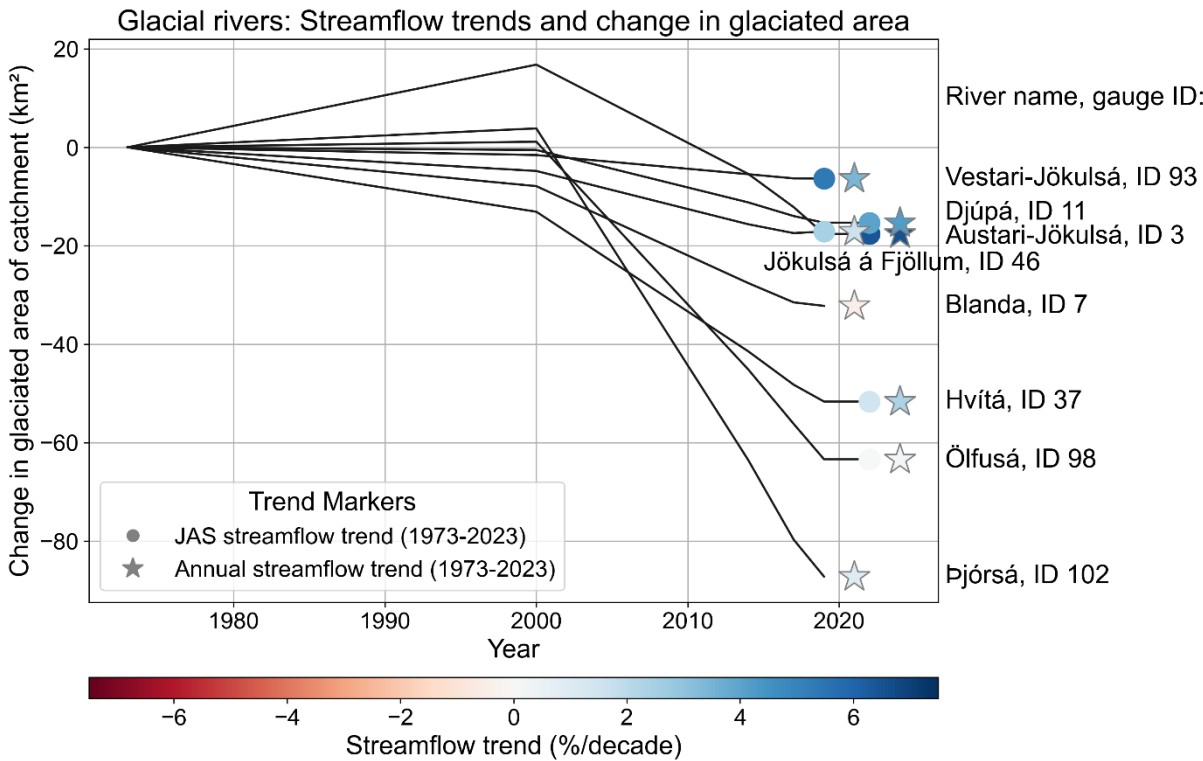

**Figure 10: Streamflow trends in glacial rivers and their relationship to changes in glaciated area from 1973 to 2023. The figure**
**highlights summer (JAS) streamflow trends and annual streamflow trends alongside changes in glaciated area, illustrating the dynamic interactions between glacial retreat and streamflow over the 50-year period.**

Figure 10 shows the relationship between streamflow trends in glacial rivers and the reduction in glaciated area from 1973 to 2023. The figure shows that rivers with minimal glacier area loss during period 1 show strong increases in streamflow, while those with more glacier area loss exhibit modest increases or even decreases. This suggests that reduced glaciated areas limit
meltwater contributions, counteracting the effects of warming temperatures on streamflow. The figure emphasizes the critical role of glacial dynamics in shaping hydrological responses to climate change.

*Period 2:*

For period 2, the analysis shows a positive trend in annual streamflow for most gauges (28 out of 37), and positive trends are more common for all seasons except summer, as for period 1. However, the number of significant trends is proportionally
lower for period 2 than for period 1; 18% of trends in annual average streamflow are significant in period 2, while 38% are significant in period 1. The seasonal trends for period 2 indicate that streamflow has increased in the fall and spring at almost all gauges, with statistically significant increases mainly in the north and east (Figure 8d). The summer season (JJA) reveals a



negative trend for most gauges (21 out of 36), although only 4 are statistically significant (Figure 8d). Most glaciated rivers have a negative trend for summer melt season (JAS) streamflow in period 2 (16 out of 17), but only 2 trends are statistically

significant (Figure 9). The three rivers draining northern Hofsjökull glacier, Blanda, Vestari-Jökulsá and Austari-Jökulsá show varying trend results, with Blanda consistently showing negative or much smaller trends than the other two (Figure S20 c and d in the Supplement). This highlights the effects of local characteristics of the catchments on trend results.

To better understand the influence of changes in evapotranspiration on streamflow, and the main drivers of streamflow changes in the summer, Figure S18 in the Supplement shows a comparison between trends in precipitation and ET trends. Annually,

precipitation increases exceed ET increases in both periods 1 and 2. Since ET in Iceland is energy-limited rather than moisture-limited (Helgason and Nijssen, 2024a), no clear relationship is observed between increases in precipitation and ET. In period 1, summer precipitation decreases, and ET increases are evident, suggesting that both factors contribute to streamflow reductions. In period 2, however, ET changes are minimal, aligning with small summer temperature trends, while precipitation decreases are more pronounced, indicating that streamflow reductions are primarily driven by precipitation decreases.

**3.3.2 Sub-seasonal trends in streamflow**

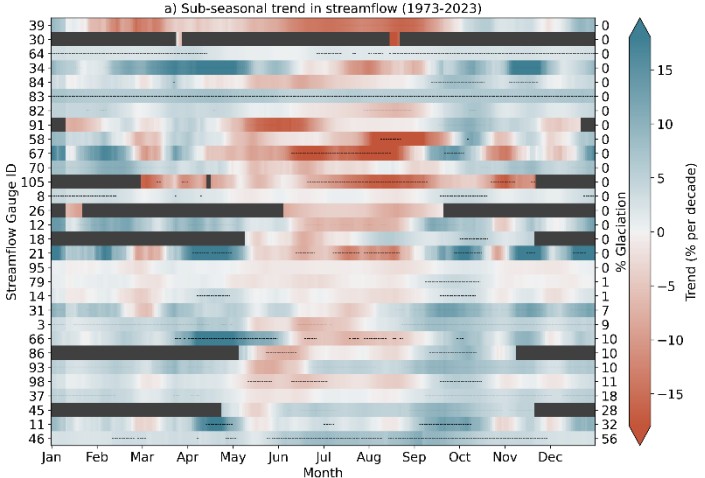



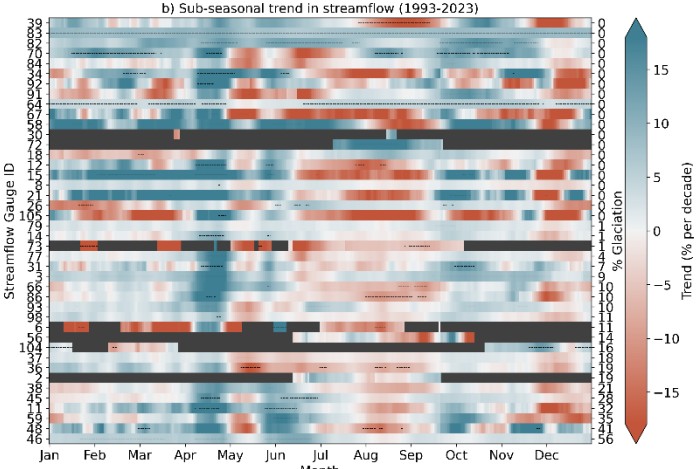

**Figure 11: Sub-seasonal trend in streamflow for the periods 1973-2023 (a) and 1993-2023 (b). Colors show the trend in 21-day centered rolling mean streamflow for each day of the year. For periods of the year with insufficient data available, a dark gray color** 390 **is shown. An overlying dashed black line indicates that the trend is significant. Left Y-axis labels show the streamflow gauge ID number. Right Y-labels show the glaciated fraction of the catchment.**

Figure 11 shows sub-seasonal trends in streamflow for periods 1 and 2. The figure shows that for period 1, significant trends (highlighted by dashed black lines) are observed for most gauges for some part of the year, with notable increases observed in winter, spring and fall for most catchments (as also shown in Figure 8b). Negative trends are predominant during the summer

season in most catchments, although positive summer trends are observed in catchments with higher glaciated fractions. Figure 11b, representing the period 1993-2023, exhibits a more pronounced pattern of frequent switches between positive and negative trends, with significant trends being fewer and more scattered throughout the year. Strong increases are particularly evident in April, followed by a decrease in May, indicating a shift in the timing of snowmelt. A strong increase is also evident in September to November. Decreases are observed from June through September for most non-glaciated catchments, and from

July through September for many glaciated catchments.





### 3.3.3 Trend in the timing of spring freshet onset, centroid of timing, and high/low flows

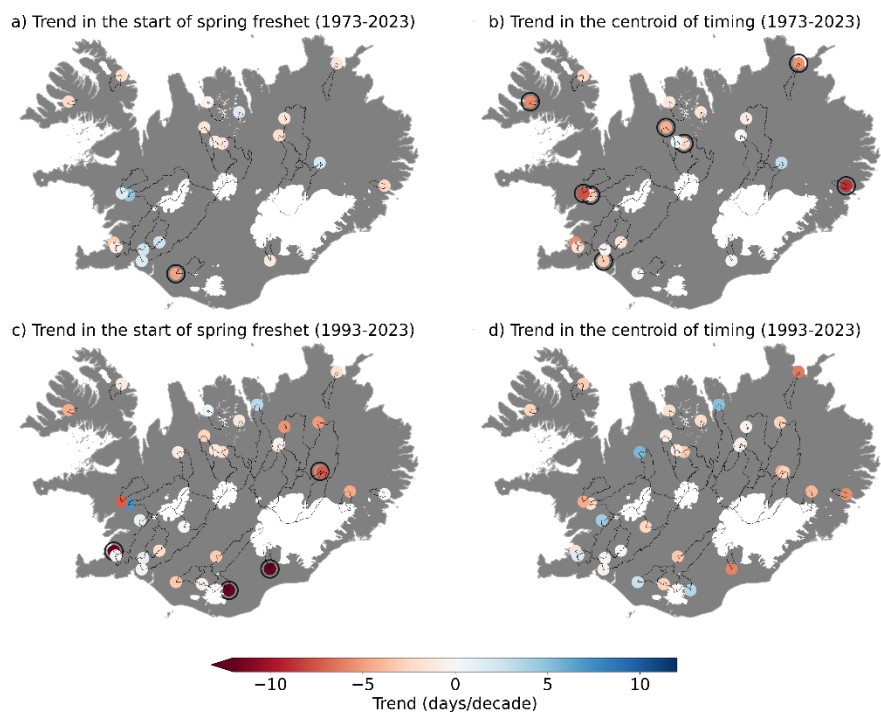

**Figure 12: Trends in the timing of the spring freshet and the centroid of timing for two periods: 1973–2023 and 1993–2023. Panels (a) and (b) show the trend in the start of spring freshet and the centroid of timing, respectively, for 1973–2023. Panels (c) and (d) display the trends for 1993–2023. Trends are expressed in days per decade, with red indicating earlier timing and blue indicating later timing. Black circles around gauge markers indicate statistically significant trends ($p < 0.05$).**

Figure 12 shows the trends in the start of spring freshet and the centroid of timing for the two periods. For period 1, the results are mixed for the start of the spring freshet (Figure 12a). Both positive and negative trends are observed across gauges, indicating variability in whether the freshet starts earlier or later. Only one gauge shows a statistically significant earlier freshet, the Eystri Rangá river in southwestern Iceland (gauge ID 14). In period 2, earlier freshets are more prevalent (Figure 12c). Four gauges show significant trends toward earlier freshets, suggesting an accelerated melt in more recent years. Figure S21 in the Supplement shows the relationship between the calculated trend in spring freshet timing, the catchment mean elevation and the catchment average trend in spring (MAM) temperature. In period 1, we see that the trend in spring freshet timing is close to 0 days per decade in all gauges. In period 2, we see that the trend is negative in most cases, although there are only 4 significant trends. At the highest elevations, we note a pattern of an increasing negative trend in spring freshet timing with higher catchment mean elevation.

For the centroid of timing (the midpoint of flow during the water year), period 1 shows mixed results (Figure 12b), with both earlier and later shifts, but 8 out of 22 gauges display significant trends toward earlier timing, which reflects the increased flow in fall season. In period 2, the results remain mixed, but no statistically significant trends are observed (Figure 12d).



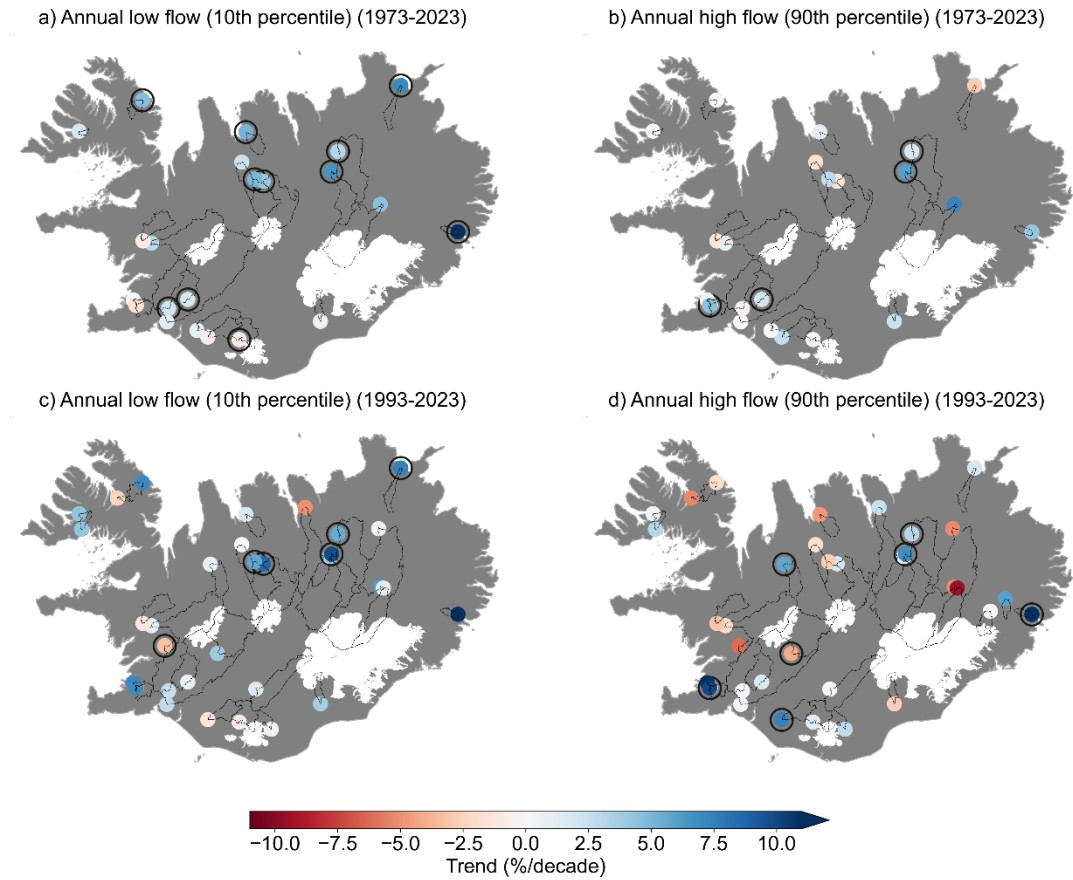

**Figure 13: Trends in annual high and low flows for period 1 and 2. Annual low (a, c) and high (b, d) flows in streamflow from 1973-2023 (a, b) and 1993-2023 (b, d). Low and high flows are defined as the 10th and 90th percentiles of annual flow. Black circles around gauge markers indicate statistically significant trends (p < 0.05). Watershed outlines are shown for each gauge.**

To investigate the change in the magnitude of low flows and floods, we extract the annual 10th and 90th percentiles of daily flow for each streamflow series. The trend in low and high flows is shown in Figure 13. For both periods, annual low flows show increasing trends for most streamflow gauges. Trends in high flows are mixed, with an almost even split between positive and negative trends for both periods (Figure 13 b and d), with fewer trends deemed significant.

## 4. Discussion

### 4.1 Multiannual variability in streamflow in Iceland

Temperature, precipitation, and streamflow in Iceland exhibit pronounced multi-annual variability, driven largely by large-scale atmospheric patterns. For example, streamflow in Iceland shows significant correlations with the Arctic Oscillation (AO) index (Figure 3). Fluctuations in streamflow reflect a complex interplay between precipitation, temperature, and glacier runoff, posing challenges for identifying long-term trends. Notably, streamflow in glaciated rivers increased significantly during the





early 2000s, coinciding with intense warming period and accelerated glacier melt. However, as the rate of warming has slowed in Iceland over the past decade, flows in most glaciated rivers have returned to near- or below-average levels. This aligns with glacier mass-balance measurements, which indicate a general slowdown in mass loss from 2011 onward compared to the rapid losses of the early 2000s (Aðalgeirsdóttir et al., 2020).

Glacier ablation has been shown to correlate with SSTs (Jónsdóttir and Uvo, 2009). Noël et al. (2022) attribute the reduced glacier mass loss after 2011 to the emergence of a regional cooling anomaly in the North Atlantic, southwest of Iceland, known as the 'Blue Blob'. This anomaly has dampened warming rates in Iceland, thereby reducing glacier meltwater runoff rates. The cause of formation of the 'Blue Blob' remains uncertain (Fan et al., 2024). Noël et al. (2022) project that the 'Blue Blob' will continue to mitigate glacier mass loss until at least the mid-2050s, based on simulations using the Regional Atmospheric Climate Model driven by the Community Earth System Model under the high-end SSP5-8.5 emission scenario. However, reliance on a single model scenario limits the robustness of such a projection, as alternative pathways could yield differing outcomes. Recent studies (Zanchettin and Rubino, 2024) document accelerated warming of SSTs in the North Atlantic in recent years, which may signal changes in the extent or influence of the 'Blue Blob' on regional climate patterns.

The 2024 water year was particularly challenging for Iceland's hydropower production. It was the driest year since 1985 and the coldest since 1998 (Figure 1), leading to significantly reduced inflows into hydropower reservoirs. Iceland's National Power Company, Landsvirkjun, has a curtailment stipulation in its agreements with its largest industrial customers, so when reservoir inflows are low, a part of the energy can be curtailed. These stipulations were activated in the 2024 water year, as well as in the following water year. Similar to the severe drought of 2013-2014, the combination of low precipitation and cold temperatures severely reduced runoff, highlighting the vulnerability of Iceland's hydropower system to prolonged dry spells and limited reservoir inflows.

Overall, the strong variability in streamflow highlights the sensitivity of Icelandic hydrology to atmospheric and glaciological conditions. The findings emphasize the importance of integrating large-scale climatic influences, such as the AO and regional North Atlantic cooling, when interpreting multiannual variability and long-term streamflow trends in Iceland. These results align closely with findings by Jónsdóttir and Uvo (2009), who reported similar variability but based on data from fewer gauges.

## 4.2 Trends in meteorological drivers and streamflow

### 4.2.1 Trends in meteorological drivers

Our results show that temperature has increased significantly in Iceland during the past decades. Spring and fall seasons show the largest positive trends during the last 30 years, while fall shows the largest positive trend during the last 50 years. Our results also show that precipitation has increased in Iceland during both periods. These results are consistent with findings from previous studies (Eythorsson et al., 2023, Björnsson et al., 2023).

An interesting finding is the increase in precipitation during September, particularly for the last 30 years. This sub-seasonal increase in precipitation has not been highlighted before to the authors' knowledge, although Gunnarsson et al. (2019) reported





a modest increasing trend in September snow cover between 2000 and 2018. These precipitation increases may reflect a shift in the timing of precipitation associated with regional atmospheric circulation changes. Even though the overall trend results were similar for the two periods, differences in the magnitude and spatial distribution of trends highlight the sensitivity of
results to the period chosen for analysis. Eythorsson et al. (2023) showed that climate simulations project similar rate increases for both temperature and precipitation during the rest of the 21$^{st}$ century.

### 4.2.2 Trends in streamflow

*Period 1: 1973-2023*

Annual streamflow trends were predominantly positive during the 50-year period (1973-2023). This increase can primarily be
attributed to rising precipitation, as well as enhanced glacier melt in glaciated catchments. The strongest positive trends were observed in the northeastern part of the country, where precipitation increases were most pronounced (Figure 4). Seasonal trends reveal that fall and winter streamflow increased significantly across most catchments, reflecting higher precipitation during these seasons and warmer temperatures. In contrast, summer streamflow showed a decreasing trend in most rivers, particularly in non-glaciated basins, which is consistent with reduced summer precipitation and higher ET.
For glaciated rivers, trends during the summer melt season were predominantly positive over the 50-year period. Enhanced meltwater contributions during this time can be attributed to strong warming trends, with the period between 1973 and 2000 being much cooler than the period after 2000, as well as effects from volcanic eruptions and glacier dynamics. For example, eruptions at Eyjafjallajökull in 2010 and Grímsvötn in 2011 deposited ash on glaciers, reducing their albedo and temporarily increasing melt rates. Similarly, the Dyngjujökull surge in 2000 expanded the glacier at lower elevations, temporarily boosting
summer flows in the Jökulsá á Fjöllum river.
Low-flow trends during 1973-2023 were consistently positive across most gauges, likely driven by enhanced winter rainfall and snowmelt contributing to groundwater recharge. High-flow trends, on the other hand, were more variable, reflecting regional differences in snowfall and extreme precipitation.

*Period 2: 1993-2023*

Trends observed over the last 30 years were more muted and variable compared to the longer period. Annual streamflow trends remained predominantly positive, but the relative number of statistically significant trends decreased. Notably, trends in glaciated rivers during the summer melt season were generally negative. Our findings suggest that in catchments with large reductions in glacier area, the summer melt season streamflow has decreased. This is likely because less glacier area results in less surface available for meltwater production. The unusually warm period in the early 2000s amplified meltwater
contributions, but this was followed by a cooler period after 2011, during which glacier melt was reduced. These cooler conditions, combined with reduced glacier area in some catchments, appear to have influenced the lack of a sustained increasing trend in the glacial rivers.
Seasonal trends during the last 30 years also differ from the longer period, with fall and spring seasons showing the most pronounced increasing trends. In non-glaciated basins, streamflow reductions in summer were more pronounced during this



period, driven by decreases in summer precipitation. The sub-seasonal trends in both periods showed streamflow increases in September and October, coinciding with enhanced autumn precipitation.

### 4.2.3 Comparisons with previous studies

Past studies of streamflow trends in Iceland reported small or insignificant trends, despite increases in precipitation (Jónsdóttir et al., 2006, 2008). Our study uses a larger dataset than has been done before, and our results show significant positive trends
in annual and seasonal streamflow in many rivers, particularly over the last 50 years. While a large majority of annual trends are positive, eight out of 25 streamflow stations show statistically significant increases in streamflow for the period 1973-2023. The same holds true for 5 out of 37 for the period 1993-2023.

Our findings align with previous studies that showed a strong correlation between streamflow in Iceland and the AO (Jónsdóttir and Uvo, 2009). We provide new insights into sub-seasonal trends and identify increases in September and October
precipitation and streamflow, which have not been reported before.

In the context of northern regions, our findings are consistent with studies that show increases in precipitation and streamflow (Box et al., 2019; Stahl et al., 2010). Many Arctic and sub-Arctic regions have observed rising winter flows due to increasing temperatures and/or precipitation (e.g. Skålevåg and Vormoor, 2021).

### 5. Conclusions

In this study, we analyzed streamflow variability and trends in Iceland, focusing on multi-annual, seasonal, and sub-seasonal changes. The results show a large inter-annual variability in streamflow and its main drivers, with notable multi-year fluctuations and a strong correlation with the Arctic Oscillation, consistent with previous studies. In recent decades, precipitation has increased in Iceland, which has led to increased annual and seasonal flows in many rivers. Positive trends are found for most gauges for both the last 30 and 50 years of annual and seasonal flows, thus marking the first study to report
such consistent results for streamflow trends in Iceland. In contrast, summer flows have declined in most non-glaciated basins, albeit insignificantly, mainly due to decreasing summer precipitation. Flow in glacial rivers was high in the early 2000s, mostly due to increased temperature-driven melt, but has decreased again in recent years, reflecting a more balanced glacier mass. Over the last 50 years, glaciated rivers exhibit positive trends in both annual and melt season flows, whereas in the last 30 years, they show negative trends during the melt season and more variable results for annual flows. Despite warming, the
timing of spring freshet has not significantly shifted earlier according to our analysis. However, the magnitude of annual low flows has increased in most rivers.

The results from this study have potential operational implications for reservoir management in Iceland. With higher expectations of increased flows during the drawdown period from fall to spring, this could allow for increased generation during this period. In addition to hydropower operations, the changing streamflow regime may affect water availability for
agricultural and ecological systems, particularly during the summer months when evapotranspiration is highest and flows are

lowest in non-glaciated rivers. Our results show that planning for water shortages in summer will be important for resource managers and policymakers.

The ERA5-Land atmospheric reanalysis data used in this study uses static glacier and vegetation masks. To better understand the drivers of changing hydrological conditions in Iceland, a hydrological reanalysis using a process-based model that
incorporates dynamic vegetation and glaciers would be useful. Such an approach would help clarify the impacts of receding glaciers, changing glacier geometries, and expanding vegetation cover on streamflow patterns. Investigating the relationship between streamflow and a broader set of climate indices, SSTs and other large-scale atmospheric patterns, could provide valuable insights into the climate drivers of streamflow variability, potentially enhancing seasonal streamflow predictions. An analysis of streamflow periodicity could reveal whether identifiable cycles influence long-term trends.

**6. Data availability**

Streamflow observations and meteorological timeseries used in this study are part of the LamaH-Ice dataset (Helgason and Nijssen, 2024b), which is available for download on HydroShare.

**7. Code availability**

The code used in this study is available on GitHub, https://github.com/hhelgason/iceland-hydro-trends

**8. Author contributions**

HBH and BN designed the study. HBH performed the data analysis and wrote the manuscript. BN, ÓGBS and AG reviewed the results and the manuscript and provided consultations and contributions throughout the work.

**9. Competing interests**

The authors declare that they have no competing interests.




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

NOAA CPC: National Oceanic and Atmospheric Administration Climate Prediction Center: Monthly Arctic Oscillation (AO)
Index, available at:
https://www.cpc.ncep.noaa.gov/products/precip/CWlink/daily_ao_index/monthly.ao.index.b50.current.ascii, and Monthly
North Atlantic Oscillation (NAO) Index, available at:
https://www.cpc.ncep.noaa.gov/products/precip/CWlink/pna/norm.nao.monthly.b5001.current.ascii, last access: November
635    25, 2024., 2024.

Noël, B., Aðalgeirsdóttir, G., Pálsson, F., Wouters, B., Lhermitte, S., Haacker, J. M., and van den Broeke, M. R.: North Atlantic
Cooling is Slowing Down Mass Loss of Icelandic Glaciers, Geophys Res Lett, 49, e2021GL095697,
https://doi.org/10.1029/2021GL095697, 2022.

Petersen, G. N. and Berber, D.: Jarðvegshitamælingar á Íslandi - Staða núverandi kerfis og framtíðarsýn (e. Soil Temperature
Measurements in Iceland - State of the Current System and Future Outlook), Icelandic Meteorological Office, Reykjavík, ISSN
1670-8261, 2018.

Pettitt, A. N.: A Non-Parametric Approach to the Change-Point Problem, J R Stat Soc Ser C Appl Stat, 28, 126–135,
https://doi.org/10.2307/2346729, 1979.

Rantanen, M., Karpechko, A. Y., Lipponen, A., Nordling, K., Hyvärinen, O., Ruosteenoja, K., Vihma, T., and Laaksonen, A.:
The Arctic has warmed nearly four times faster than the globe since 1979, Communications Earth & Environment 2022 3:1,
3, 1–10, https://doi.org/10.1038/s43247-022-00498-3, 2022.

Rawlins, M. A. and Karmalkar, A. V.: Regime shifts in Arctic terrestrial hydrology manifested from impacts of climate
warming, Cryosphere, 18, 1033–1052, https://doi.org/10.5194/TC-18-1033-2024, 2024.

Raynolds, M., Magnússon, B., Metúsalemsson, S., and Magnússon, S. H.: Warming, sheep and volcanoes: Land cover changes
in Iceland evident in satellite NDVI trends, Remote Sens (Basel), 7, 9492–9506, https://doi.org/10.3390/rs70809492, 2015.

Sen, P. K.: Estimates of the regression coefficient based on Kendall's tau, J Am Stat Assoc, 63, 1379–1389,
https://doi.org/10.2307/2285891, 1968.



Skålevåg, A. and Vormoor, K.: Daily streamflow trends in Western versus Eastern Norway and their attribution to hydro-meteorological drivers, Hydrol Process, 35, e14329, https://doi.org/10.1002/HYP.14329, 2021.

Snorrason, A.: Hydrologic variability and general circulation of the atmosphere, in: XV Nordisk Hydrologisk Konferens (NHK-90, Kalmar, Sweden, 29 July-1 Aug 1990). National Energy Authority: Reykjavík, OS90027/VOD-02, 1990.

Stahl, K., Hisdal, H., Hannaford, J., Tallaksen, L. M., Van Lanen, H. A. J., Sauquet, E., Demuth, S., Fendekova, M., and Jódar, J.: Streamflow trends in Europe: evidence from a dataset of near-natural catchments, Hydrol. Earth Syst. Sci, 14, 2367–2382, https://doi.org/10.5194/hess-14-2367-2010, 2010.

Theil, H.: A rank-invariant method of linear and polynomial regression analysis. I, II, III, Nederl. Akad. Wetensch., Proc., 53, 386–392, 521–525, 1397–1412, 1950.

Thompson, D. W. J. and Wallace, J. M.: The Arctic oscillation signature in the wintertime geopotential height and temperature fields, Geophys Res Lett, 25, 1297–1300, https://doi.org/10.1029/98GL00950, 1998.

Wanner, H., Brönnimann, S., Casty, C., Gyalistras, D., Luterbacher, J., Schmutz, C., Stephenson, D. B., and Xoplaki, E.: North
Atlantic oscillation - Concepts and studies, Surv Geophys, 22, 321–381, https://doi.org/10.1023/A:1014217317898/METRICS, 2001.

WMO: World Meteorological Organization: Manual on Low-flow Estimation and Prediction. Operational Hydrology Report No. 50. WMO-No. 1029. ISBN 978-92-63-11029-9, 2008.

Yue, S., Pilon, P., Phinney, B., and Cavadias, G.: The influence of autocorrelation on the ability to detect trend in hydrological
series, Hydrol Process, 16, 1807–1829, https://doi.org/10.1002/HYP.1095, 2002.

Zanchettin, D. and Rubino, A.: Accelerated North Atlantic surface warming reshapes the Atlantic Multidecadal Variability, Communications Earth & Environment 2024 5:1, 5, 1–10, https://doi.org/10.1038/s43247-024-01804-x, 2024.

Zaqout, T. and Andradóttir, H. Ó.: Impact of climate change on soil frost and future winter flooding, in: Novatech 2023 : 11e Conférence internationale sur l'eau dans la ville, 2023.

Zaqout, T., Andradóttir, H. Ó., and Sörensen, J.: Trends in soil frost formation in a warming maritime climate and the impacts on urban flood risk, J Hydrol (Amst), 617, 128978, https://doi.org/10.1016/J.JHYDROL.2022.128978, 2023.