# Peer review of "Understanding Changes in Iceland's Streamflow Dynamics in Response to Climate Change"

_EGUsphere, 2024_

## Author Comment (AC1)

Manuscript egusphere-2024-4186: Response to reviewer #1
Hörður B. Helgason, Andri Gunnarsson, Óli G. B. Sveinsson and Bart Nijssen, Understanding
Changes in Iceland's Streamflow Dynamics in Response to Climate Change

Original reviews in black
**Responses in blue and bold**

**RC1: Comment on egusphere-2024-4186 by Referee #1:**
**Manuscript Review: "Understanding Changes in Iceland's Streamflow Dynamics in Response to Climate Change"**

**1) Impact**

This study makes a significant contribution to our understanding of how climate change affects Icelandic streamflow dynamics. By utilizing the extensive LamaH-Ice dataset, the authors provide valuable insights into long-term hydrological trends in Iceland. The findings have important implications for hydropower management, water resource planning, and ecological sustainability. Moreover, the regional focus on Iceland enriches the global discussion on climate-induced hydrological changes.

**2) Strengths**

Comprehensive Data Utilization: The use of the LamaH-Ice dataset, which covers a broad network of largely undisturbed catchments, increases the reliability of the study.

Multi-Decadal Analysis: The examination of streamflow trends over both 30- and 50-year periods allows for a nuanced understanding of short- and long-term hydrological changes.

Climatic Correlations: The study effectively links streamflow variations with large-scale climate drivers such as the Arctic Oscillation (AO) and North Atlantic Oscillation (NAO), which strengthens the analysis.

Operational Relevance: The discussion on hydropower implications makes the study practically useful for policymakers and energy managers.

Clear Visualization: The figures—including maps, heatmaps, and rolling mean analyses—effectively convey trends and spatial variations.

**3) Weaknesses**

Limited Discussion on Anthropogenic Influence: Although the study excludes heavily regulated rivers, it does not sufficiently explore how human interventions (e.g., land use changes, hydropower infrastructure) might interact with climate-driven changes. In particular, the catchment of the Kárahnjúkar Hydropower Plant is scarcely considered because the paper focuses on gauging stations where the river is minimally affected by human activities. But wouldn't the effect of a changing stream flow be particularly interesting for Iceland's biggest hydropower plant?

**The primary focus of this study is to assess natural hydrological changes driven by climate change, and therefore, anthropogenic influences such as land use changes and hydropower infrastructure are beyond the scope of this analysis. We will clarify the study's scope in the introduction to explicitly state that this research examines climate-driven streamflow trends in near-natural catchments and does not aim to assess anthropogenic influences on streamflow.**

Uncertainty in Precipitation Data: The reliance on ERA5-Land reanalysis for precipitation introduces potential biases, as noted by the authors. A discussion on alternative precipitation datasets or validation techniques could strengthen the results.

**We discussed the uncertainty with and biases in the precipitation of the ERA-5 Land reanalysis in the original Lamah-Ice paper (Helgason and Nijssen, 2024). A large part of these biases stemmed from an underestimation of orographically enhanced precipitation along the Icelandic coast. We recognize the limitations associated with using ERA5-Land reanalysis data for precipitation trends. We will perform the same trend analysis for precipitation from the CARRA reanalysis (Schyberg et al., 2020), to assess similarities and differences between the two datasets.**

Glacial Dynamics Interpretation: The study links decreasing glacial river flow trends over the past 30 years to glacier retreat, but it does not explore potential non-linear meltwater contributions or threshold effects. In large glaciers such as Vatnajökull, enhanced ice melt may play a more dominant role than glacier retreat in influencing meltwater contributions. For instance, Figure 5 shows the precipitation trend, which correlates well with results from Kárahnjúkar watershed (Heger et al. 2025) where contributions to streamflow increase in spring and autumn while snowmelt decreases. Specifically, Figure 5 indicates that in the region of Kárahnjúkar, spring precipitation has increased by approximately 10%, whereas winter precipitation has decreased.

**We appreciate the reviewer's comment on non-linear meltwater contributions and potential threshold effects in glacial streamflow trends. We acknowledge that the relationship between glacier response to climate change and streamflow changes is complex and varies between different glacier types. The presence of both mountain glaciers and large ice caps further complicates this relationship due to differences in meltwater storage and dynamics. We will add this to the discussion in the manuscript.**

Limited Policy Discussion: Although the manuscript mentions implications for hydropower management, it does not propose specific adaptive strategies.

**We will propose adaptive strategies for hydropower management in the revised manuscript.**

Statistical Significe: Figure 6 illustrates sub-seasonal trends in temperature and precipitation and shows that in the second period analyzed, the trends are considerably stronger, which could be interpreted as an intensification of extremes. The authors mention that many trends are not statistically significant, a point that is reflected in the data. Some variables decrease between 1973 and 2023 but then increase again between 1993 and 2023. Additionally, glacier melt was less intense in the last decade compared to the 1990s, reflecting not only variability in annual weather but not necessarily a robust trend. This variability could be discussed in the context of the uncertainty of a weakening of the Atlantic Meridional Overturning Circulation (AMOC) (see Rahmstorf 2024), which may influence both climate extremes and glacier dynamics.

**We appreciate this suggestion and agree that discussing the observed variability in trends within the context of potential drivers, including the weakening of the Atlantic Meridional Overturning Circulation (AMOC), will strengthen the manuscript. We discuss this in the revised manuscript and reference relevant studies, such as Rahmstorf (2024), to provide additional context on how AMOC variability may influence climate extremes and glacier dynamics.**

**4) Specific Editorial Suggestions**

Line 80          "it's location"    "its location" (remove the apostrophe)

Line 115          "which only returns as runoff up to decades later": Consider rewording for clarity: "which contributes to runoff decades later"

Line 199          "The warming appears to have slowed in recent years.": Consider adding a reference or supporting data for this claim

Line 390          "An overlying dashed black line indicates that the trend is significant ($p < 0.05$)." Consider rewording to match the style of other trend significance descriptions

Line 414          "We see that the trend is negative in most cases, although there are only 4 significant trends.": Suggest quantifying "most cases"

Line 505          "While a large majority of annual trends are positive...": Consider rewording for clarity: e.g. "Although most annual trends are positive, only eight out of 25 stations show statistically significant increases for 1973-2023."

**We appreciate the reviewer's detailed editorial suggestions and will incorporate the revisions to improve clarity and consistency.**

**Final Recommendations**

To enhance the impact of the conclusions, the authors could emphasize some key quantitative findings (please check the numeric values):

Temperature Rise: Annual average temperatures have increased by approximately 0.2°C per decade.

Precipitation Increase: Total precipitation rises by about 1.5% per decade, with notable seasonal variations (around a 10% increase in spring(?) and decreases in winter).

Streamflow Variability: While 21 out of 25 gauges show positive streamflow trends for 1973–2023, glaciated rivers display predominantly negative summer trends in the recent 30-year period— this could possibly be linked to a weakening of the AMOC (Rahmstorf 2024)?

Highlighting quantitative results in the conclusions would strengthen the paper's data-driven arguments and improve its relevance for climate impact assessments and policy formulation.

Address some of the identified weaknesses: Expand the discussion on anthropogenic influences (perhaps the watershed of Kárahnjúkar Hydropower Plant could be used as a representative example), address uncertainties in precipitation data by comparing with alternative datasets or validation techniques, and consider non-linear responses in glacier melt.

Enhance policy relevance: suggest specific adaptive strategies for hydropower and water management to improve the applied value of the study.

Correct Editorial Errors: Implement the minor editorial corrections listed above to enhance clarity and precision throughout the manuscript.

**In the updated manuscript we will address all these valuable recommendations as outlined above, except for an analysis of anthropogenic influence on streamflow since this is outside of the scope of this study.**

**References**

**Schyberg, H., Yang, X., Køltzow, M. A. Ø., Amstrup, B., Bakketun, Å., Bazile, E., Bojarova, J., Box, J. E., Dahlgren, P., Hagelin, S., Homleid, M., Horányi, A., Høyer, J., Johansson, Å., Killie,**

M. A., Körnich, H., Le Moigne, P., Lindskog, M., Manninen, T., Nielsen Englyst, P., Nielsen, K. P., Olsson, E., Palmason, B., Peralta Aros, C., Randriamampianina, R., Samuelsson, P., Stappers, R., Støylen, E., Thorsteinsson, S., Valkonen, T., and Wang, Z. Q.: Arctic regional reanalysis on single levels from 1991 to present, Copernicus Climate Change Service (C3S) Climate Data Store (CDS), https://doi.org/10.24381/cds.713858f6 (last access: 15 February 2023), 2020.

---

## Author Comment (AC2)

Manuscript egusphere-2024-4186: Response to reviewer #2
Hörður B. Helgason, Andri Gunnarsson, Óli G. B. Sveinsson and Bart Nijssen, Understanding Changes in Iceland's Streamflow Dynamics in Response to Climate Change

Original reviews in black
**Responses in blue and bold**

**RC2: Comment on egusphere-2024-4186 by Referee #2:**
**General comments**

The paper addresses observed changes and trends in streamflow on Iceland and explores their meteorological drivers, i.e., temperature, precipitation. It starts with analysing the annual streamflow variability and its link to atmospheric circulations (NAO and AO), followed by a detailed analysis of trends in annual, seasonal and smoothed daily flow for 37 streamflow stations across Iceland. The dataset includes both glaciated and non-glaciated catchments for two separate, but overlapping periods. As such it provides a valuable assessment of changes in observed streamflow and meteorological drivers in a cold-climate region sensitive to global warming. The recently published LamaH-Ice dataset for Iceland provides in this context a unique opportunity to study such changes, representing a region often left out in large-scale hydrological studies. The paper is in parts well written.

The main aim is to "investigate the multi-annual variability in Iceland's climate and analyse changes in streamflow, aiming to link these changes to shifts in hydro-meteorological drivers and changes in glacier extent and mass". The latter part "changes in glacier extent and mass" is not explicitly analysed in the paper through own analyses, rather the results are discussed in light of previously reported changes in glacier extent and mass. This in contradiction to shifts in hydro-meteorological drivers, which are quantified as trends, although the link to changes in streamflow is discussed in qualitative terms. This distinction should be made clear to the reader from the start.

**While our study does not directly analyze glacier mass balance, we do compare trends in annual and summer melt-season streamflow to changes in glacier extent (Figure 10). Beyond this, glacier extent changes and mass loss are currently discussed in the context of previously documented trends rather than through independent calculations. We will make this clear to the reader from the start by making the following changes to the text:** *"[…] investigate the multi-annual variability in Iceland's climate and analyze changes in streamflow, aiming to link these changes to shifts in hydro-meteorological drivers and changes in glacier extent."*

The study is thorough in that it investigates trends in streamflow and its meteorological drivers for different periods, temporal resolution and variables. The analysis is straightforward, using well-established methods. It is descriptive in its presentation, with an extensive result section, listing the results in a sequential way supported by a series of informative figures. As such, it resembles more a scientific report than a research paper. The results are discussed in a qualitative way, comparing the streamflow trends with trends in precipitation and temperature. The reasons for choices taken are not always well motivated; for example, what is the purpose/added value of looking at the daily trend in smoothed 21- day averages? And why is the spring freshet, rather than the timing of the peak flow (often used in the literature), chosen (or why not both)? Choosing indices commonly used in the literature, facilitates comparison with other studies. The paper would have benefitted from being tightened somewhat with a clear motivation for why - and how - the different analysis contributes to the aim and research question defined.

**We acknowledge that the results section is currently extensive. To enhance clarity and focus, we will streamline this section by emphasizing the most critical findings that directly address our research questions. We will relocate detailed analyses (e.g. Figures 6 and 11 that show the sub-**

seasonal trends in meteorological forcings and streamflow for all catchments) to the supplementary material. Since Figure 7 effectively summarizes these trends for meteorological forcings, we believe it is sufficient for the main text.

We will revise the introduction to explicitly articulate the motivation for including sub-seasonal analyses and provide relevant citations. While annual and seasonal trend analyses are valuable for identifying long-term hydrological changes, they may overlook important shifts occurring at finer temporal scales. Sub-seasonal trends can provide valuable insights even when no significant annual or seasonal trends are detected. Moreover, when seasonal trends are present, this analysis helps pinpoint when changes occur in greater detail—for example, the increase in precipitation in September, as shown in Figure 7.

For the spring freshet timing, our choice was based on its hydrological significance in cold regions, where the onset of snowmelt-driven runoff is a key indicator of climate-driven changes in seasonal hydrology. However, we recognize that peak flow timing is also commonly used in the literature, and we will assess peak flow timing as well and compare the results.

The abstract states "We then analyze trends during the last 30 and 50 years (actually 31 and 51 years given that the start and end year are included) in annual, seasonal, and smoothed daily streamflow volumes, the timing of the spring freshet, and extreme flow conditions, linking these changes to environmental conditions and catchment attributes." This latter is not entirely true as only three catchment attributes are listed (Table S3), one being the degree of anthropogenic impact (classified into four groups), which is used merely to exclude (heavy influenced) catchments from the study, not in the analysis itself. Of the other two, only the percentage of glaciers is included in the analysis and discussion, whereas BFI is not used at all. Accordingly, a quantitative analysis on how these attributes relate to the trends detected is missing. The streamflow series stem from a dataset (LamaH-Ice), which includes a wide range of catchments attributes, thus this is a missed opportunity. Including such an analysis would have enhanced the relevance and quality of the paper, allowing an in-depth analysis of the patterns found.

Our analysis is based on hydrological years, spanning October 1, 1973, and October 1, 1993, to September 30, 2023, which correspond to 50 and 30 complete years, respectively. We will clarify this in the manuscript to avoid ambiguity. We appreciate the reviewer's suggestion regarding the inclusion of additional catchment attributes in the analysis. While we acknowledge the potential value of further catchment attribute analysis, we aim to maintain a focused discussion. Therefore, we will assess whether integrating select attributes - such as runoff ratio, depth to bedrock, or dominant land cover class - adds meaningful insights to our findings without overly extending the results section.

The exception being a separate analysis found in the Supplement. Figure S21 nicely shows the relationship between the calculated trend in spring freshet timing, the catchment mean elevation and the catchment average trend in spring (MAM) temperature. It is here concluded (period 2) that there is "a pattern of an increasing negative trend in spring freshet timing with higher catchment mean elevation". Is this so? Figure S21 shows that the largest negative trends in the spring freshet timing (approx. -18 days/decade) is found at the lowest elevation (< 200m asl). These results are in either case not discussed or commented on later in the paper.

We appreciate the reviewer's careful assessment of Figure S21 and their attention to detail. The figure illustrates the relationship between spring freshet timing, catchment mean elevation, and spring (MAM) temperature trends. In the manuscript, we stated: *"At the highest elevations, we note a pattern of an increasing negative trend in spring freshet timing with higher catchment mean elevation."* This observation was based on the seven catchments with mean elevations above 800 m a.s.l. We acknowledge that the small number of high-elevation catchments does not provide sufficient evidence to support a general conclusion of an increasing negative trend in spring freshet timing with higher elevation.

**A more accurate observation is that there is no clear evidence that catchments at lower elevations experience systematically earlier freshet timing compared to higher-elevation catchments. We have made the following changes to the text starting in line 415: "* There is no clear evidence that catchments at lower elevations experience systematically earlier freshet timing compared to higher-elevation catchments.*"**

The discussion section is nicely written summarising key findings of the study, albeit rather brief when it comes to discussing the results as such. It would have benefitted from an overall broader discussion, including a more extensive section on how the findings compare to previous studies (from Iceland or comparable sites). Section 4.2.3 mentions the lack of other studies on Iceland, and refers to two publications (from 2006 and 2008). However, the European study by Blöschl et al., 2017 (mentioned in the Introduction) and the Nordic study by Wilson et al., 2010 (not in the reference list) both include Iceland (there may be others). How do the findings from these studies compare to the results of this study? It may be valuable to compare also with studies from similar regions in other parts of the world where glaciers are present (Section 4.2.3 does not mention glacier fed rivers), i.e., do the findings from this study agree with other studies or are they unique to Iceland?

**In line 54, after the sentence: "*A key takeaway was that despite a substantial increase in precipitation, no corresponding increase was found in the flow of non-glacial rivers (Jónsdóttir et al., 2006).*"**

**we have added:**

**"*Wilson et al. (2010) analyzed trends in annual and seasonal streamflow in Icelandic rivers for 1961–2000 and found no significant trends in either annual or seasonal streamflow, confirming the findings by Jónsdóttir et al. (2006). Their analysis also showed no shift in the timing of the spring floods.*"**

**In the discussion section (line 507), we have added:**

**"*Our analysis shows that the timing of the spring freshet has not systematically shifted, consistent with the findings of Wilson et al. (2010) for the period 1961–2000.*"**

**Blöschl et al. (2017) found that the timing of annual maximum floods in southwestern Iceland shifted later in the year during the period 1960–2010, while in northeastern Iceland, flood timing remained stable or occurred earlier, as noted in the introduction. Once we have completed our analysis of annual peak flow timing, we will compare our findings with theirs.**

**To maintain focus and conciseness, we will not expand the discussion to include comparisons with studies from other glaciated regions.**

The paper has an extensive Supplement assessing the Homogeneity of streamflow records (S1), Changes in evapotranspiration compared to changes in precipitation (S2), Trends in streamflow (S3) and Overview of gauges used in the study (S4). Most of this material are not considered necessary to include (see suggestions below).

**In Conclusion**: The paper is for reasons stated above along with the many issues raised regarding the data and methodology as elaborated below, not recommended for publication as it reads now. Rather, a resubmission can be considered for review. Accordingly, detailed textual comments are not provided at this stage.

**Issues that need clarification/discussion**

1. **Presentation of the dataset:**

-It is stated that 38 stations (later reduced to 37 due to the homogeneity test) were used, however, in Section 3.2 one refers to the whole dataset of 107 catchments in the LamaH-Ice dataset, including human influenced catchments, and in the Data section reference is made to 79 stations (Section 2.3). This is confusing.

**We analyze trends in meteorological forcings using data from all 107 catchments in the LamaH-Ice dataset. For streamflow trends, we use 25 gauges for period 1 and 37 gauges for period 2 (following the homogeneity test). We acknowledge that the current wording may be unclear, and we will revise Section 3.2 to explicitly state the number of stations used in each case to avoid confusion.**

- Are any of the catchments analysed sub-catchments of larger catchments included?

**A few of the catchments included in our analysis are sub-catchments of larger catchments that are also analyzed, though this is relatively uncommon. This can be observed in the figures displaying basin outlines along with gauge locations (e.g., Figures 3, 8, 12, and 13). To ensure clarity, we will explicitly state this in the Data section.**

- The analysis is done for two different periods; 1973-2023 and 1993-2023, 31 and 51 years respectively). It would have been interesting to know the mean streamflow, precipitation and temperature for the two periods.

**We will add the mean streamflow, precipitation, and temperature for the two analyzed periods to Table S3 to provide additional context.**

- In Figure 1 the period 1951-2024 is used; why is this period chosen? Later in the text, the year 2024 is highlighted as a particular dry year and has been given a separate paragraph in the discussion. Why this focus on 2024 if not included in the trend analysis?

**The period 1951–2024 was chosen primarily based on data availability from the ERA5-Land reanalysis. Initially, ERA5-Land was available from January 1, 1950, but it has since been extended back to 1940. In the updated manuscript we will make sure to use either the period 1950-10-01 to 2024-09-30 or 1940-10-01 to 2024-09-30. The year 2024 is highlighted separately in the discussion due to its extreme conditions, which underscore the high variability in precipitation and temperature in Iceland. Although 2024 is not included in the trend analysis due to the lack of streamflow data for that year, its recent occurrence makes it particularly relevant for water management and stakeholders in Iceland. We will clarify this distinction in the revised manuscript.**

- Figure 1 further includes a rolling 5-year mean and trendlines; how do these trendlines relate to the trends later calculated?

**The trendlines in Figure 1 provide a visual representation of overall long-term changes, while the formal trend analysis later in the manuscript is based on different time periods. We will explain this difference.**

- In the first part of the study (annual variability) the hydrological year is used; is this also the case for the second part of the study on trends?

**The hydrological year is used for both annual variability and trend analysis. We will confirm this explicitly in the text.**

- The seasonal study includes four commonly used seasons where summer is defined as June, July, August. Later, another summer season is introduced, the i.e., July, August, September also referred to as the summer melt season, as being more relevant for assessing the peak contribution from glaciers. This choice ought to be well introduced earlier (now it appears for

the first time in the Result section) and discussed in terms of interpreting the results of the two different 'summer' periods.

**The introduction of two different summer definitions (JJA and JAS) will be clarified earlier in the text, along with the rationale for considering JAS as a more relevant period for glacier melt contributions.**

- The area of the catchments should be provided as it is important for evaluating the spatial variability in meteorological variables, notable in large catchments where a mixed trend pattern may impact the catchment average trend signal

**We will include catchment areas in Table S3 in the Supplement to help assess spatial variability in meteorological trends.**

- Uncertainty is not discussed, neither in terms of the observed data (streamflow) nor in the gridded ERA5-Land variables (temperature and precipitation), and how this may impact the results

**We recognize the need to discuss data uncertainties. We will add a section addressing uncertainties in observed streamflow and ERA5-Land variables and their potential impacts on results.**

- It is stated that precipitation in the ERA5-Land reanalysis is underestimated for Iceland (Helgason and Nijssen, 2024a); what about temperature? Cryosphere processes are very sensitive to air temperature, notable the zero-degree crossing.

**To address this, we will assess potential temperature biases in ERA5-Land over Iceland by comparing its temperature estimates with in situ weather station measurements and other reanalysis products. This analysis will help evaluate the reliability of ERA5-Land for applications involving cryospheric and hydrological processes.**

- How does ERA5-Land perform for other glaciated areas of the world?

**While ERA5-Land's global performance is an interesting question, our study is focused specifically on Icelandic conditions.**

- Has the ERA5-Land dataset been evaluated for its ability to reproduce trends in observations for Iceland? If not, it is recommended to include an at-site comparison where observed (station) precipitation (P) and temperature (T) time series are compared with the ERA5-Land grid cell representing the location of the stations

**ERA5-Land has not been explicitly evaluated for trend reproduction in Iceland. This has been done for temperature in other areas, e.g. Turkey** (Yilmaz, 2023) **and China** (Zhao & He, 2022)**. We will include a comparison of observed station-based temperature time series with ERA5-Land data. Precipitation observations are difficult to use for this kind of comparison due to snow and wind in winters. However, we can compare trends in precipitation between ERA5-Land and other reanalysis datasets (e.g. CARRA).**

- ET values are likely rather low on Iceland, a small change in ET (in mm), may become a notable percentage change (add to the discussion).

**Evapotranspiration (ET) values are indeed low in Iceland. Due to this, even small absolute changes in ET (measured in mm) can correspond to significant relative percentage changes. This is why we report ET trends as a percentage of the average watershed precipitation per decade in Figure 4. We will emphasize this point further in the discussion.**

- It is suggested to present spatial trend patterns in the meteorological variables for the whole of Iceland. By mapping the trends for each grid cell (ERA5-Land dataset) over Iceland, one can detect potentially regional diverging trend pattern that may help understand the spatial aggregated trends in streamflow for large catchments (as they may cover an area with mixed trend patterns).

**We analyze trends for all 107 catchments in LamaH-Ice, providing high spatial coverage across Iceland. Given the consistency of trends observed across regions (see Figures 4 and 5), a gridded analysis would not add substantial new insights. Our approach ensures that catchment-scale trends, which are most relevant for streamflow analysis, are well represented.**

**2. Supplementary material**

The material consists of four sections, of these:

- S1 is not needed (sufficient to state that the series were tested for homogeneity using a specific methodology);

**Section S1 will be removed, as the homogeneity test methodology is sufficiently described in the main text.**

- S2 is partly repeating what is presented in the main text, basically using another unit and is thus not needed. The exception being trend in evapotranspiration (ET), which in Fig. 4 is presented in % of precipitation/decade, which makes it difficult to distinguish the role of precipitation (P) vs ET. In Figure S17, the map legend states "mm/decade" for ET, whereas in the Figure caption it says that ET trends are shown as "percentage of annual precipitation per decade". What is correct?

**In Figure S17, "mm/decade" is the correct unit. In Figure 4, ET trends are expressed as a percentage of average annual precipitation per decade, as ET values are low, making even small absolute changes result in significant relative percentage changes. This approach also provides useful context in relation to precipitation trends. We will revise Section S2 to remove redundant information while retaining ET trends in mm/decade for clarity. Additionally, we will correct the discrepancy in Figure S17 to ensure consistency between the map legend and the figure caption.**

- S3 also provides material that can be deleted, including Fig. S19, Table S1 and S2, whereas Figure S20 and S21 may be kept (in the Supplementary or integrated in the main text).

**We agree that section S3 can be reduced. Figures S19 and Tables S1–S2 will be removed, while Figures S20 and S21 will be retained or integrated into the main text.**

- S4 and Table S3 should be kept, but revised (replacing BFI with e.g., catchment area if BFI is not used).

**Section S4 and Table S3 will be retained but revised to replace BFI with catchment area, since the BFI is only used in the homogeneity analysis which will be removed.**

**3. Aggregating climate trends across the catchment area**

The location of each gauging station is shown by a circle on the map and its colour (red, blue and white) indicates whether a negative, positive or no trend is observed at the site. Trends in

catchment average climate variables, such as temperature and precipitation, are represented in a similar way, i.e., by a circle at the gauging station. This raises a few questions:

- How were the catchment average climate variables derived from the gridded ERA-5Land? **Catchment-average climate variables were derived from ERA5-Land by computing the area-weighted arithmetic mean of grid cell values within each catchment boundary, including those that partially overlap it. This is further described in the LamaH-Ice data description paper (Helgason and Nijssen, 2024).**

- How was the catchment average trend calculated from the gridded dataset?

**The catchment average trend was calculated from the catchment-average climate variable timeseries. We will clarify this in section 2.3 of the manuscript.**

4. **Calculation and discussion of trends**

- In the calculation of trend in spring freshet and centroid of timing, missing data are filled for gaps of 60 days or less, implying that there can be periods with more than 10% of daily data per year (= 36 days) or season (= 9 days) missing that are included. 10% is earlier stated as the limit for including a year in the time series of annual or seasonal trends (Section 2.3). How does this agree?

**Years with more than 10% missing daily data are excluded from annual and seasonal streamflow trend analyses. This criterion also applies to the analysis of the onset of spring freshet and centroid of timing. However, for these metrics, gaps of up to 60 consecutive days are filled beforehand, which may indeed result in the inclusion of years with more than 10% missing data. This approach ensures that as many time series as possible are retained. We acknowledge this inconsistency and will revise the manuscript to clarify the criteria for handling missing data across all analyses.**

- Further, if more than 10% are missing, that year is excluded from the trend analysis. How is a missing year dealt with in the time series; just skipped or indicated as missing? Is the assessment of significance adjusted accordingly?

**When a given year (or season) has more than 10% missing daily data, we exclude that year from the analysis. This ensures that only years with sufficiently complete records are used to represent the typical flow conditions. We omit series with more than 20% of annual/seasonal values missing. A missing year is dropped from the series before calculating trend and significance. We acknowledge that omitting these years reduces the total number of observations and may affect the trend estimates and the significance test's power.**

- Have the annual values been tested for autocorrelation? If the time series contain autocorrelation for the annual time step, this will violate the trend calculation (refer findings in Wilson et al., 2010).

**Yes, the annual streamflow values were tested for autocorrelation, revealing presence of autocorrelation in some series. We therefore use a modified mann-kendall test that adjusts the variance of the test statistic to account for autocorrelation. See first paragraph of Sect. 2.6.**

- The trend direction is shown as blue and red. Is there a range in z-values for when a trend is defined as zero?

**We do not use a range in z-values for zero trends.**

- Suggest to focus on discussing significant trends and only comment on nonsignificant trends if there are a clear regional or temporal pattern, which then contains information beyond the individual station.

**Our manuscript already follows this approach for meteorological trends, where we primarily discuss significant trends and comment on non-significant trends only when they exhibit a clear regional or temporal pattern. To ensure consistency across all variables, we will refine the results and discussion sections to apply this approach systematically to streamflow trends as well.**

- Figure 10 shows an interesting plot of streamflow trends in glaciated rivers only. But why is the trend not calculated for both the annual and JAS season for all rivers (circles are missing for some of the rivers)?

**The reason for omitting JAS season trends is due to upstream reservoirs that influence the natural flow in the river. Annual trends are included since the reservoirs do not significantly affect annual streamflow volumes. We will clarify this in the figure caption.**

- Low flow and high flow are represented by the 10th and 90th percentile from the whole time series, and no distinction is made between seasons. As the dominating low flow or high flow season may be in summer or winter (or both), events generated by different processes may be included in a given time series, which hampers a discussion on the causes of the trend seen.

**We will distinguish seasonal contributions to high and low flows to improve the interpretation of percentile-based streamflow trends.**

**5. Streamflow anomalies**

In Figure 2, streamflow anomalies are expressed as deviations from the long-term mean (in percentage). However, it is stated that the calculations are done based on the available data for each station, which imply that the results are not comparable across stations given that these are of different length. To allow comparison, the same reference period must be used to derive the mean for each station (a common period), following which deviation from the mean can be extended beyond the reference period.

**We acknowledge that using the full available record for each station results in non-comparable anomaly calculations. We will standardize the reference period across stations to ensure consistency.**

**6. Sub-seasonal (daily) trends**

One may argue that a choice of a 30 (or 31) day moving window may be better than a 21-day window, as it resembles the monthly time resolution (this is also the time window used for gap-filling). A miss a discussion on the cause of the sharp, and consistent, transition between positive and negative trends seen in Fig. 6. The overlying dashed black line is hard to see in the figures.

**We tested multiple window lengths and found that a 21-day window effectively balances the need for temporal precision while maintaining robustness in the analysis. A discussion will be added on the sharp transitions between positive and negative trends observed in Figure 6 (e.g. in September/October and December). We will also enhance the visibility of the dashed black line indicating trend significance.**

**7. Glacier catchments**

Trends in streamflow are presented for all catchments in the main body of text, whereas the results for glacier catchments are presented in the Supplement (S3). Why not present the results for non-glaciated catchments as well?

**Figure S20 in the Supplement presents trends in annual and summer melt-season streamflow, specifically for glacier-fed rivers. This figure facilitates a direct comparison between annual and melt-season trends in glacial rivers. In contrast, the main manuscript presents the trends in annual and summer streamflow for all gauges, ensuring a broader overview.**

**Consistency in terminology**:

Both watershed (US) and catchment (UK) are used (choose one)

Both fall (US) and autumn (UK) are used (choose one)

**Both terms are understood and accepted in both countries and widely used in English around the world. The interchangeable use of these terms does not lead to confusion and we therefore see no need to change this.**

**References:**

Yilmaz, M. (2023). Accuracy assessment of temperature trends from ERA5 and ERA5-Land. *Science of The Total Environment*, *856*, 159182. https://doi.org/10.1016/J.SCITOTENV.2022.159182

Zhao, P., & He, Z. (2022). A First Evaluation of ERA5-Land Reanalysis Temperature Product Over the Chinese Qilian Mountains. *Frontiers in Earth Science*, *10*, 907730. https://doi.org/10.3389/FEART.2022.907730/BIBTEX